**Registered Report**

# Information transfer within and between autistic and non-autistic people

Catherine J. Crompton ●[1] ✉, Sarah J. Foster[2], Charlotte E. H. Wilks[3], Michelle Dodd[1], Themis N. Efthimiou[1], Danielle Ropar[3], Noah J. Sasson[2], Martin Lages ●[4] & Sue Fletcher-Watson ●[5]

Autism is clinically defined by social communication deficits, suggesting that autistic people may be less effective at sharing information, particularly with one another. However, recent research indicates that neurotype mismatches, rather than autism itself, degrade information sharing. Here, using the diffusion chain method, we examined information transfer in autistic, non-autistic and mixed-neurotype chains ($N = 311$), replicating and extending a key study. We hypothesized that information transfer would deteriorate faster and rapport would be lower in mixed-neurotype compared with single-neurotype chains. Additionally, we examined whether informing participants of the diagnostic status of their chain and whether information was fictional or factual impacted performance and rapport. We found no difference in information transfer between single-neurotype and mixed-neurotype chains. Non-autistic chains indicated higher rapport, and disclosing diagnosis improved rapport. This result challenges assumptions about autistic communication deficits but contrasts with prior findings. Enhanced participant heterogeneity and methodological differences may explain these unexpected results. Protocol registration The Stage 1 protocol for this Registered Report was accepted in principle on 23 August 2022. The protocol, as accepted by the journal, can be found at https://osf.io/us9c7/.

The diagnostic criteria for autism spectrum disorder (henceforth 'autism')[1] include lifelong impairments in social communication and interaction that contribute to broad social disabilities and poor functional outcomes[2]. These difficulties are associated with fewer friendships[3], less social support[4], loneliness[5], challenges securing and maintaining employment[6–8], poorer mental health[9,10] and reduced quality of life[11].

The majority of research so far has assumed a deficit model of autism, characterizing the differences in autistic sociability and communication as deviations from normality in need of remediation (for a review, see ref. 12). This model, however, ignores the relational nature of social interaction and locates the cause of social interaction difficulties

exclusively within the autistic person[13]. A growing body of research has begun to examine the factors that influence how autistic people interact with non-autistic people and the impact that this has on autistic social experiences[14,15]. Communication is bidirectional, and social difficulties experienced by autistic people can be exacerbated by the behaviours, social judgements and misunderstandings of non-autistic social partners[13,16,17].

Though attitudes about autism are improving[18], stigma remains high[19–22]. Non-autistic people form rapid negative judgements about autistic people on the basis of on their non-normative social presentations[14] that are strongly associated with a greater reluctance to interact with them[14,15]. Such judgements reduce the social opportunities

[1]Centre for Clinical Brain Sciences, University of Edinburgh, Edinburgh, UK. [2]School of Behavioral and Brain Sciences, The University of Texas at Dallas, Richardson, TX, USA. [3]School of Psychology, University of Nottingham, Nottingham, UK. [4]School of Psychology and Neuroscience, University of Glasgow, Glasgow, UK. [5]Salvesen Mindroom Research Centre, Centre for Clinical Brain Sciences, University of Edinburgh, Edinburgh, UK. ✉e-mail: catherine.crompton@ed.ac.uk

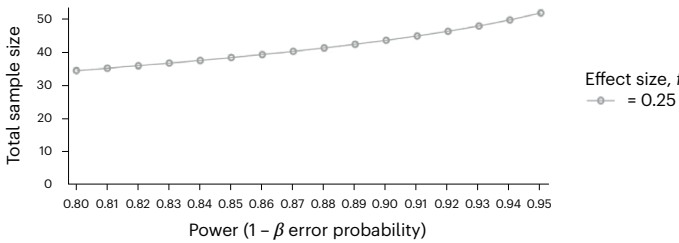

**Fig. 1 | Visualization of the G\*Power analysis.** A priori power analysis (G\*Power)[99] on the original data from Crompton et al.[34] using a mixed design (chain type–between and position–within) with the number of chains as the sample size unit. *F* tests from repeat-measures ANOVA, within–between interaction. *N* groups = 3, *N* measurements = 6, correlation among repeated measures of 0.4, non-sphericity correction $\varepsilon$ = 0.7, $\alpha$ error probability of and effect size *f* = 0.25.

afforded to autistic people by the non-autistic majority and present barriers to achieving personal and professional goals[23,24].

Non-autistic people also misconstrue and misunderstand what autistic people are thinking and feeling[25]. Countless studies have documented the difficulties of autistic people when accessing and interpreting the mental states and emotion of non-autistic people[26,27], yet the reverse occurs as well. Non-autistic people struggle to identify autistic facial expressions[28] and understand autistic mental states[29], and these misunderstandings are associated with liking them less[30]. Non-autistic people also overestimate how egocentric autistic people are[31] and even report being helpful to autistic people in circumstances when, objectively, no evidence of helpful behaviour is demonstrated[32].

Unsurprisingly given this context, autistic people report feeling more comfortable and relaxed in the company of other autistic people[33–37]. They are more likely to want to spend time with other autistic people[16,38], disclose more to autistic interaction partners[38] and empathize more with other autistic individuals[39]. Autistic people also often communicate in a more direct, literal or 'frank' manner that is sometimes interpreted as rude by non-autistic people but is welcomed as a preferred communication style by other autistic people[14,33]. Other recent studies have provided further support for these conclusions. Autistic interactions feature non-normative social behaviours that result in enhanced communication with other autistic people[32,36]. Autistic people also use common social cues differently when communicating with autistic and non-autistic people[40], and the positive rapport produced between autistic pairs extends beyond self-report and can be detected by observers[41]. These findings challenge assumptions that impaired social communication and connection are inherent to autism and suggest that the social and communication difficulties of autistic people arise, at least in part, from a mismatch between autistic and non-autistic modes of communication and understanding. This is supported by empirical studies of the kinematic dissimilarity hypothesis of social interaction, which suggests that kinematic similarity is important for action predication and social interaction[42]. There are well-documented kinematic differences between autistic and non-autistic people[43], and recent studies have found that autistic observers are more able to accurately predict autistic actions than non-autistic actions[44], and conversely that non-autistic observers are more able to accurately predict non-autistic actions than autistic actions[42]. These data suggest that difficulties in social interactions arise because of objective differences in the way that autistic and non-autistic people communicate, rather than an autistic social deficit.

This literature, however, is still in its infancy and, up to this point, has consisted of single studies using relatively small samples and divergent methodologies. More rigorous examination of autistic-to-autistic communication, using larger samples and standardized methodology across independent sites, is needed. Crompton et al.[34] provided the most direct and ecologically valid evidence of intact communication

efficacy between autistic partners, with selective breakdowns in communication occurring between autistic and non-autistic people. Using a cultural learning paradigm to capture the transmission of information between autistic–autistic pairs, non-autistic control pairs and mixed (that is, autistic and non-autistic) pairs, Crompton et al.[34] found that autistic partnerships facilitate interaction for autistic people. Specifically, (1) autistic people transfer information to and from other autistic people as effectively as non-autistic people do with each other, (2) the quality of information sharing selectively breaks down when one person is autistic and the other is not and (3) interpersonal rapport is higher within than between diagnostic groups and these feelings accompany information-sharing benefits.

Although this study has been impactful and has generated new lines of enquiry, it was generated from a relatively modest sample (*N* = 72) recruited from a single geographical location. The reproducibility and generalizability of experimental results is an essential part of the scientific process and is particularly important in studies producing results that conflict with prevailing assumptions—and indeed in this case with clinical criteria. The aim of the proposed study is to replicate the work of Crompton et al.[34] but with a substantially increased sample size (see Figs. 1 and 2) across three independent sites, two of which are independent of the initial study. This study examines the effect of matched and mismatched neurotype on information transfer and self-reported rapport between autistic and non-autistic people. The procedure for participation is presented in Table 1. We predict a replication of the original finding showing that information transfer degrades most rapidly in chains of mixed (autistic with non-autistic interactions), compared with chains of matched-pair interactions (H1; see Table 2 for full details). We also predict that rapport will follow a similar pattern, with enhanced rapport in matched-neurotype pairs and poorer rapport in mixed pairs (H2; see Table 2 for full details).

The current study also extends beyond ref. 34 in two important ways. First, we explore whether being informed of the diagnostic status of one's interaction partner affects information transfer. In ref. 34, participants were informed of the diagnostic status of their partner before their interaction. However, in real-world interactions, many autistic adults choose not to disclose their diagnosis to avoid bias and discrimination[45], and emerging evidence suggests that awareness of an autism diagnosis influences how autistic people are perceived by non-autistic people, but not by other autistic people[16,46]. A diagnostic-informing manipulation in the current study offers an opportunity to explore whether such patterns have downstream effects on communication and rapport. We also explored whether

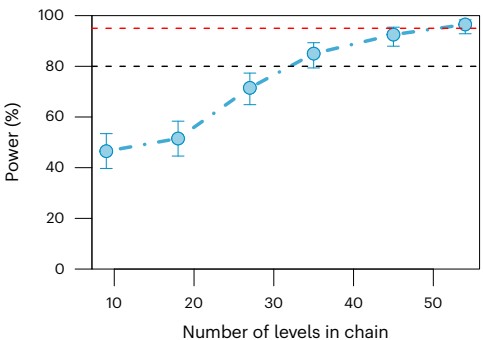

**Fig. 2 | Visualization of the results from the power analysis based on simulations.** Simulation results (R package simr[83]) for the number of chains ranging from 9 to 54 to determine the exact power for the interaction effect 'chain type–position' (blue circles) and 90% CIs (error bars). The linear mixed-effect model 'Story_prop - chain type × position + (1|chain)' was fitted to the original data from Crompton et al.[34] with three chain types and six positions. The estimated coefficients were reduced by one-third to make sample size estimation more conservative. The horizontal dashed black and red lines indicate 80% and 95% power, respectively.

**Table 1 | Procedure for participation**

| Online, in advance of the in-person participation | | |
|---|---|---|
| Completed independently by the participant | Information sheet and providing informed consent | |
| | Participant demographic questionnaire | |
| | RAADS[70]/ RAADS-14[72] | |
| **In-person participation** | | |
| Administered by researcher | WASI-II[80] (if not previously completed on enrolment to participant database) | |
| Completed as part of a diffusion chain | Diffusion chain task 1 (fictional or factual depending on order) | |
| Completed independently by participant | Rapport measure 1 | |
| Completed as part of a diffusion chain | Diffusion chain task 2 (fictional or factual depending on order) | |
| Completed independently by participant | Rapport measure 2 | |

the efficacy of information transfer differs as a function of material content. Reference 34 involved participants retelling a fictional story to partners sequentially in a chain, and it is unclear whether the findings would differ between fictional and factual passages matched on length and complexity. Prior research has indicated that autistic people have greater interest in, and facility with, information about factual systems[47], but whether this translates to efficacy of communication and rapport is unclear.

## Results
### Information transfer
Separate analyses of task performance for fictional and factual information transfer were conducted in R[48]. The following hypotheses were investigated: hypothesis 1a: participants in autistic and non-autistic chains will share significantly more details than in mixed chains (main effect 'chain type'); hypothesis 1b: information will deteriorate between participants (order '1–6') in each chain (main effect 'order'); hypothesis 1c: information decay will be steeper in mixed chains compared with autistic and non-autistic chains while autistic and non-autistic chains will perform similarly (interaction between 'chain type' and 'order').

Multilevel mixed-effect models (MLMs; R package lme4 (ref. 49), all models presented in lme4 syntax) considered dependencies between observations within chains by estimating random intercepts and random slopes for each chain. Covariate order was replaced by log-transformed log(order) to capture nonlinear decay of information within diffusion chains. To assess site differences, models included the control variable site (Dallas, Edinburgh or Nottingham), and to assess any impact of participants being informed of the diagnostic status of their chain, we included diagnostic informing (info for short; informed or uninformed). A third variable first (fictional story first or factual story first) addresses the counterbalanced sequence of tasks. Full details of effects of site, diagnostic informing and task order are detailed in Supplementary Information. Bayes factors (BFs) were established by comparing a simpler model without a predictor and associated interactions against a more complex model with the predictor and associated interactions (generalTestBF function in the R package BayesFactor using default priors and repeated measures for observations in each chain)[50,51]. The story scores are shown in Fig. 3.

**Fictional information.** A model comparison in terms of the Akaike information criterion (AIC) favoured the model ~chain type × log(order) + site × info × first + (1 + log(order)|number), which also met the criterion for multi-collinearity (variance inflation factor (VIF) <5).

Alternative models with more interactions gave similar results but improved model fit only marginally and increased collinearity (VIF >5). This model predicted task performance (range 0–30) with an adjusted (marginal) $R^2 = 0.55$ and adjusted (conditional) $R^2 = 0.87$ (R package MuMIn[52]). Satterthwaite's method was used to adjust degrees of freedom in $t$-tests (R package lmerTest[53]).

As predicted, the main effect of log(order) was large and statistically significant ($b = -5.29$, $t(51.3) = -9.07$, $P < 0.001$, $\eta^2 = 0.17$, 95% confidence interval (CI) −6.43 to −4.15; R package r2glmm[54]). There was no statistically significant difference of task performance for chain type, meaning no significant differences between autistic, non-autistic and mixed chains (BF of 7.91 for the simpler model). A significant but unpredicted interaction between chain type 'autistic' and log(order) reduced task performance compared with chain type 'non-autistic' ($b = -1.73$ (s.e.m. of 0.83), $t(48.9) = -2.09$, $P = 0.042$, $\eta^2 = 0.01$, 95% CI −3.37 to −0.11). Keeping participants uninformed about diagnostic status had negligible effects (BF of 0.85 for the simpler model).

**Factual information.** A model comparison in terms of AIC favoured a full MLM featuring all predictors and interactions but indicated multi-collinearity (VIF >5). Model ~chain type × log(order) + site × in fo × first + (1 + log(order)|number) was a more parsimonious model in terms of AIC and collinearity (VIF <5). This model predicted performance scores with an adjusted (marginal) $R^2 = 0.60$ and adjusted (conditional) $R^2 = 0.89$. As predicted, the main effect of log(order) was strong and statistically significant ($b = -4.57$, $t(50.9) = -8.29$, $P < 0.001$, $\eta^2 = 0.26$, 95% CI −5.65 to −3.49). There was no statistically significant difference of task performance for chain type (BF of 125.0 for the simpler model). Keeping participants uninformed about diagnostic status had negligible effects (BF of 0.41 for the simpler model).

**Summary.** H1a was not confirmed, as no significant effect between autistic, non-autistic and mixed chains emerged. H1b was confirmed, showing a strong effect of (log-transformed) order on information decay. H1c was not confirmed, as there was no interaction between the mixed chain type and (log-transformed) order, as predicted, although an unpredicted significant interaction between autistic chain type and (log) order was observed. As groups significantly differed on intelligence quotient (IQ), gender and ethnicity, these control variables were included in post hoc analyses. The effects were small and did not affect the results described above (Supplementary Information). Additional models using linear and nonlinear approaches confirmed that the findings above were robust (Supplementary Information).

### Self-rated rapport
The following hypotheses for self-rated rapport were investigated, with 'teachers' (transmitting information in a chain) and 'learners' (receiving information in a chain) analysed and reported separately: hypothesis 2a: there will be higher rapport scores in autistic and non-autistic chains compared with mixed chains (main effect of 'chain type'); hypothesis 2b: the rapport score will increase in the second relative to the first transfer between participants in mixed chains (interaction between 'chain type' and 'first'); hypothesis 2c: participants in the informed condition will have higher rapport scores than participants in the uninformed condition (main effect diagnostic informing or 'info' for short). The rapport results are shown in Fig. 4.

**Teacher rapport.** MLM with random intercepts for each chain were used, with Satterthwaite's method to adjust degrees of freedom in $t$-tests (R package MuMIn[52]). A model comparison for teacher rapport scores in terms of AIC favoured a model where ~chain type + story type × first + info + site + (1|number) predicted the rapport scores (range 0–500). More complex models with higher-order interactions improved model fits only marginally and increased collinearity (VIF >5). This model predicted the teacher rapport score with an adjusted

## Table 2 | Design table

| Question | Hypothesis | Sampling plan (for example, power analysis) | Analysis plan | Interpretation given to different outcomes |
|---|---|---|---|---|
| (1) Do autistic, non-autistic and mixed chains differ on transfer of fictional information, when participants are aware of the diagnostic status of their partner? | (1a) Autistic and non-autistic chains will share significantly more details than mixed chains (null: no difference between chain types). (1b) Information will deteriorate between participant position (1–6) in each chain, and this will occur earlier in the mixed chains compared with the non-autistic chains. Autistic and non-autistic chains will not differ significantly (null: no difference between positions). (1c) Deterioration may occur significantly earlier in mixed chains compared with autistic and non-autistic chains (null: no difference between chain types across conditions). | As described in the 'Power analysis' section, a priori power analysis for a linear model with mixed design based on the original study[34] to test hypothesis 1 suggests a sample size of 54 chains/324 participants to achieve a power of 95%. | A linear regression with mixed design, with the dependent variable of individual participant score and independent variables of chain type (autistic, non-autistic or mixed) and chain position (1–6) will be performed. This replicates the analysis used in the original study. | We anticipate main effects of chain type and position and an interaction between group and chain position (1–6), with the mixed group showing a steeper decline in number of details recalled across the chain compared with the autistic and non-autistic conditions. If we find a significant effect of chain type with the mixed chains having lower scores across participant position, then this will be evidence in support of H1a. Additionally, if we find an interaction between chain type and position that mirrors the original effect, then this will be evidence in support of H1c. If we do not find a significant main effect of chain type, we will reject our hypothesis. Alternative patterns of results may include (1) a selective pattern of more rapid information loss in autistic chains indicating an autistic disadvantage in information transfer in a task or, conversely, (2) a selective pattern of less rapid information loss in non-autistic chains, indicating a non-autistic advantage in information transfer in this task. |
| (2) Are there differences in interpersonal rapport between autistic, non-autistic and mixed groups, and does this vary when diagnosis is known or between first and second diffusion chains? | (2a) We predict significantly higher rapport in the autistic and non-autistic chains relative to the mixed chain (null: no difference between the groups). (2b) Given that contact with autistic people is associated with more favourable impressions, less stigma and more inclusionary attitudes[20,46], we hypothesize that rapport will increase in the second relative to the first interaction between partners in mixed chains (null: no difference between first and second interactions). (2c) Participants in the informed condition will have higher rapport than participants in the uninformed condition (null: no difference between informed and uninformed conditions). | As outlined in the 'Power analysis' section, this study has been powered for all hypotheses on information loss. The relatively large sample size should allow us to also examine our second hypotheses with 95% power because effect sizes in the mixed design had similar size. | We will perform a linear regression with the dependent variable of individual rapport score and predictor variables chain type (autistic, non-autistic or mixed), interaction order (first or second) and diagnostic informing (informed or uninformed). We will then create more complex models by adding one predictor at a time. In a comprehensive model comparison using information criteria, we will compare simpler with more complex models that include additional variables as well as two-way interactions to determine the most parsimonious model. We will only interpret the effects of the most parsimonious model. | Evidence of a difference between the autistic, non-autistic and mixed chain types on rapport would suggest that interpersonal rapport is experienced differently depending on the context of the interaction (single diagnosis/mixed diagnosis groups). A lack of evidence of difference between the autistic, non-autistic and mixed chain types on rapport would suggest that interpersonal rapport does not depend on the diagnosis or participants (autistic/non-autistic) and/or the context of the interaction (single diagnosis/mixed diagnosis groups). Evidence of an improved model fit due to other variables including participant role, diagnostic informing, task type or interaction order would suggest that these factors have a significant impact on rapport. If this is the case, we will examine the direction of this effect (and possible interactions). An interaction between chain type and other factors would suggest that being aware of the diagnostic status of an interactive partner impacts rapport more profoundly in one or more groups. |

(marginal) pseudo $R^2 = 0.07$ and adjusted (conditional) pseudo $R^2 = 0.10$ (R package MuMIn[52]), indicating that random intercepts captured some variability. Teachers in non-autistic chains had significantly higher rapport scores than those in mixed chains ($b = -35.8$, $t(46.9) = -3.45$, $P = 0.0012$ $\eta^2 = 0.03$, 95% CI $-56.2$ to $-15.4$) and in autistic chains ($b = -33.3$, $t(54.3) = -3.21$, $P = 0.002$, $\eta^2 = 0.03$, 95% CI $-53.7$ to $-12.9$). Teachers who were uninformed about the neurotype of the learner had lower rapport score than teachers who were informed but the effect did not reach significance ($b = -17.8$ (s.e.m. of 8.98), $t(47.4) = -1.98$, $P = 0.054$, $\eta^2 = 0.01$, 95% CI $-35.4$ to $-0.20$). The effect of story type (BF of 2.47 for the simpler model) and first (BF of 0.48 for the simpler model) was negligible ($\eta^2 < 0.01$), but the interaction between story type and first was significant ($b = -36.5$, $t(458.9) = -2.48$, $P = 0.013$, $\eta^2 = 0.01$, 95% CI $-65.3$ to $-7.69$), where rapport was higher for the second (factual) task when the fictional task was first. No other significant effects were noted.

**Learner rapport.** The same model as above predicted learner rapport score with an adjusted pseudo (marginal) $R^2 = 0.082$ and adjusted

(conditional) pseudo $R^2 = 0.16$ (R package MuMIn[52]). Learners in non-autistic chains had higher rapport than those in mixed chains ($b = -22.3$, $t(47.7) = -2.01$, $P = 0.04$, $\eta^2 = 0.01$, 95% CI $-43.7$ to $-0.94$), but not significantly higher than in autistic chains ($b = -20.8$, $t(47.6) = -1.92$, $P = 0.06$, $\eta^2 = 0.01$, 95% CI $-42.0$ to $0.37$). Learners who were uninformed about the neurotype of the teacher had a significantly lower rapport score compared with learners who were informed ($b = -19.8$, $t(47.4) = -2.11$, $P = 0.04$, $\eta^2 = 0.02$, 95% CI $-38.2$ to $-1.43$). The main effect of story type (BF of 0.02 for the simpler model) and first (BF of 0.01 for the simpler model) was negligible ($\eta^2 < 0.01$), but the interaction between story type and first was significant ($b = 50.0$, $t(457.1) = 3.87$, $P < 0.001$, $\eta^2 = 0.03$, 95% CI $24.7$ to $75.3$), where rapport was higher for the second (factual) task when the fictional task was first. No other statistically significant effects were observed.

**Summary.** H2a was partially confirmed: the difference between non-autistic and mixed chains was statistically significant, but the difference between autistic and mixed chains was negligible. H2b was not confirmed: there was no significant difference between rapport

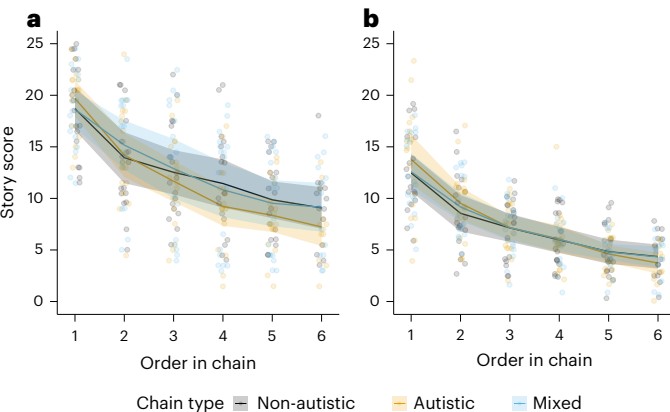

**Fig. 3 | Story scores across chain types and chain order. a,** The mean fictional story scores (±1 s.e.m.) reported by six successive participants across three chain types: non-autistic, autistic and mixed. **b,** The mean factual story scores (±1 s.e.m.) for the same chain types. Individual data points (n = 311) represent raw story scores for each participant, with colours corresponding to chain types. To improve clarity, data points are jittered horizontally. Both **a** and **b** reveal a general decline in story scores with increasing chain order, with variations in the rate of decline observed across chain types.

scores when participants completed them after the first and second diffusion chain tasks. H2c was partially confirmed because reductions of rapport scores for uninformed compared with informed participants were small, and the effect was significant for the rapport scores for learners but not for teachers. The most robust effects were reduced teacher and learner rapport scores for mixed and autistic chains, as well as the interaction between story type and first. Post hoc analyses that included IQ, gender and ethnicity as control variables did not affect the results described above (Supplementary Information).

## Discussion

Universal deficits in social communication, including behaviours used for social interaction, a reduced interest in peers and 'abnormal social approaches', are central to the current diagnostic criteria for autism[2]. It follows that autistic people should demonstrate impairments in sharing information with others. However, previous research has indicated that autistic and non-autistic people are similarly accurate at sharing information with others of the same neurotype, with both selectively experiencing poorer information transfer in mixed-neurotype settings[34]. The current study rigorously tested the generalizability of these findings in a much larger and more diverse sample.

An important feature in the original study also appeared in this replication: there was no significant difference in information transfer accuracy between autistic and non-autistic chains, indicating that autistic adults are not impaired relative to non-autistic participants in their ability to transfer information accurately within same-neurotype chains. This holds true for different samples and sites, and is inconsistent with the core deficit theory of autism[55], current diagnostic criteria[2] and a canon of social-cognitive autism research[56] that assumes that autistic people should perform more poorly at communicative tasks than non-autistic people, regardless of context. These findings validate the presence of effective communicative skills in autistic people, even if they do not necessarily adhere to non-autistic social norms.

However, mixed-neurotype chains also did not differ from either autistic or non-autistic single-neurotype chains in information transfer. In other words, while autistic interaction pairs showed equal communicative abilities when compared with non-autistic pairs, we did not find the predicted selective breakdown of communication when autistic and non-autistic people were interacting. The pattern occurred in both fictional and factual tasks and strongly favoured a model without chain types.

As previous research indicates that disclosing diagnosis results in more positive impressions of autistic people[46], it was hypothesized that participants who were informed of the diagnostic status of their partner would experience higher rapport than uninformed participants (H2c). The results elicited a mixed picture. Across the board, rapport was higher when diagnosis was disclosed, aligning with previous work[46]. When participants rated rapport as someone sharing information (that is, 'teacher'), non-autistic chains had higher scores than autistic and mixed chains. Autistic people particularly enjoyed talking to other autistic people. However, when participants rated rapport as someone learning information (that is, 'learner'), autistic and non-autistic chains did not differ, and both had higher scores than mixed chains. People preferred learning from someone of the same neurotype.

While some findings from the original study generalized, others did not. One likely explanation for results differing between the two studies is the increased heterogeneity of the sample. The participants in the current study encompassed a range of ages, genders and ethnicities. The autistic group specifically had a wide range of IQ scores and included both clinically and self-identified participants who reported a range of ages of (self-)diagnosis. Additionally, participants were drawn from multiple recruitment routes, at three sites across three nations—albeit all Western cultures. The original study had far less diversity in all these respects. While our questions focus on the impact of neurotype on information transfer, other individual differences such as age[57] and gender[58] also shape human interactions. For example, matching on gender and ethnicity can lead to improved learning[59] and rapport[57,58,60]. In this study, increasing the diversity of the sample may have influenced interactions, overshadowing an effect of neurotype matching. Nonetheless, variability—including variability of IQ—did not influence the autistic sample from sharing information and experiencing rapport with each other. In summary, we infer that it is not simply whether someone is autistic or not that determines the success of their interaction with others, but also the match or mismatch of other characteristics. Disentangling such intersectional effects would require systematically manipulating multiple demographic variables, which was beyond the scope of the current study.

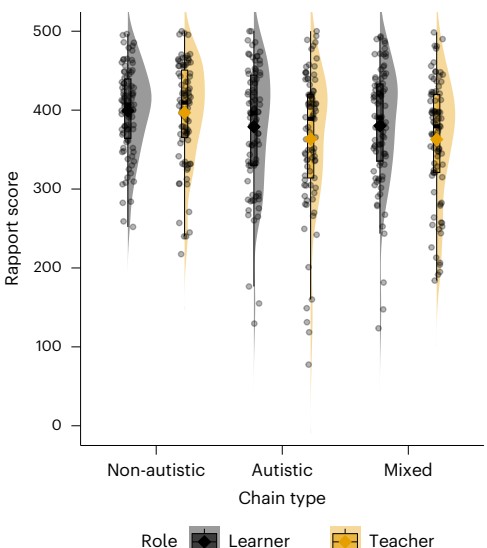

**Fig. 4 | Violin and box plots illustrating the rapport score distributions by chain type and role.** The violin plots (shaded transparently) show the distribution of rapport scores across chain types and roles (learner in grey and teacher in orange) (n = 311). The box plots (outlined in black) display the median (centre line), interquartile range (25th to 75th percentiles), and whiskers extending to 1.5 times the interquartile range. The jittered grey points represent individual data values, and diamonds indicate mean scores for each group.

This study has limitations, which could be addressed by future research. First, all participants were drawn from the USA and UK, and thus were aligned with Western cultural and communicative norms. We do not know whether similar findings would arise across participants from different cultural backgrounds. Second, the mean IQ of the participant groups were high, and participants all engaged in verbal communication; it is not known whether a similar pattern of findings would occur with participants with an intellectual disability or with participants communicating using non-speaking channels. Finally, although we examined individual effects of gender and ethnicity on performance, diffusion chains included participants of different and ethnic backgrounds, meaning that we could not examine the effects of matching participants on these other important variables to examine their potential effects on communication, information transfer and rapport.

Although this study was a conceptual replication of ref. 34, it included several methodological differences, yet none should have differentially affected one chain type more than any other. First, we wrote a new fictional story to avoid the risk that participants read the published original before taking part. The original and replication stories were carefully matched on length and reading difficulty. Second, to standardize administration across sites, participants watched a video of a man recounting the fictional story to them, whereas in the original study the fictional story was recounted live by a female researcher. There is no evidence of differences in recall between face-to-face and video instruction in non-autistic populations[61]. Although no research has examined this for autistic people, this methodological feature is unlikely to explain the pattern of effects already discussed—that is, absence of differences between chain types. Third, in this study, half the participants heard the factual narrative first, whereas the fictional task was the only task in the original study. Again, even if some kind of practice effect arose from hearing a factual story first, there is no discernible reason why this should affect chains differentially. Finally, this study used six-person chains, to avoid collecting data influenced by floor effects at later positions. As effects were detectable after just six positions in the original study, this change is unlikely to have affected the results reported here.

Future research could examine the content of the interactions to investigate whether participants in the three conditions differ in the types of information they share or the way it is conveyed. One could also ask whether these findings replicate across different types of tasks—motor as well as verbal domains, or in open-ended, creative tasks. One further question concerns the extent to which the success of mixed-neurotype interactions depends on pre-existing beliefs about being autistic. An autistic person who is working hard to fit in (often referred to as 'masking'[62]) could end up more effectively sharing information with non-autistic partners, and both give and receive higher rapport scores. Extrapolating beyond our experimental scenario, a pattern like this could deliver short-term interactional gains at the expense of long-term mental health[63,64]. Conversely, it is possible that positive beliefs about being autistic could negatively affect interactions across neurotypes. This might happen if autistic people are working to be their authentic selves, rather than effortfully bridging the communication gap, and/or if they bring an expectation of lack of success to the interaction. In the original study, participants were older than in the current study, and autistic people have reported masking less as they get older[65]: it may be that the younger participants were more likely to mask, inflating cross-neurotype information transfer and rapport scores. Future work should consider including a scale of masking, such as the Camouflaging Autistic Traits Questionnaire[66], to ascertain its potential role.

## Conclusions

This well-powered, preregistered study replicated the finding that autistic and non-autistic people share information and establish rapport with similar levels of success within same-neurotype contexts. Additionally,

no difference was found in performance in mixed-neurotype chains. A growing body of empirical evidence[32,34,35], along with first-person accounts from autistic people[13,67–69], have shown a preference for same-neurotype interactions, with mixed-neurotype interactions being more challenging to navigate. The experimental context tested here may have failed to capture difficulties experienced in real-world cross-neurotype interactions. This could be due to real-world conversations being more dynamic and interactive than the unidirectional information transfer tasks used here. Research examining the role of multiple intersecting identities is needed, but for now, these data support a growing challenge to the lack of contextual nuance in the diagnostic criteria for autism.

## Methods

### Ethics

This study was carried out in accordance with the British Psychological Society's Code on Human Research Ethics and the American Psychological Association's Ethical Principles of Psychologists and Code of Conduct. Experimental procedures were reviewed and approved by the University of Edinburgh's Medical Research Ethics Committee, the University of Nottingham School of Psychology Ethics Committee and the University of Texas at Dallas's Institutional Review Board. All participants provided written informed consent before participating and were remunerated for their time (£30/US$40).

### Pilot data

Data demonstrating the feasibility of our approach were collected and published in ref. 34. This pilot study used a diffusion chain paradigm with autistic, non-autistic and mixed chains, and assessed the fidelity of information transfer of a fictional story and between-participant rapport in each condition. Thus, we were confident in the feasibility of the methods and proposed analyses for this study.

In the pilot study, a standard linear regression model including predictor variables of chain type and position (order) showed a steeper decline for the mixed chains of alternating autistic and non-autistic participants ($\beta = -6.04$ (s.e.m. of 1.32), $P < 0.0001$), with chains consisting solely of autistic or non-autistic participants not differing significantly ($\beta = 0.13$ (s.e.m. of 1.32), $P = 0.93$). This conclusion was also supported by additional analyses showing a significant interaction between the chain type and position, indicating a significantly faster deterioration rate of information sharing in the mixed chains ($\beta = 0.57$ (s.e.m. of 0.26), $P < 0.05$). Together, chain type and position accounted for 85% of the variance in the amount of information shared ($F(5,66) = 77.05$, $P < 0.0001$, $R^2 = 0.85$). Thus, autistic and non-autistic chain types did not differ in their information-sharing capabilities, and penalties for information transfer selectively occurred in the mixed chains where diagnostic status was mismatched. In the current study, we suggested a revised data analysis for the power calculations (see the 'Sampling plan' section).

### Design

**Experimental design.** This study used a mixed experimental design incorporating both between and within-groups factors. Between-groups factors included chain type (autistic, non-autistic or mixed) and diagnostic informing (informed or uninformed), with task type (fictional or factual) as within-group factors.

**Participants and recruitment.** Three hundred twenty-four adult participants (162 autistic people and 162 non-autistic people) comparable on age, gender and IQ were recruited across three sites (the University of Edinburgh, the University of Nottingham and the University of Texas at Dallas), and 311 (154 autistic people and 157 non-autistic people) attended research days. The autistic group was predominantly female (51.3%) and non-binary (30.52%) with a mean age of 28.68 years (s.d. 11.18 years). The non-autistic group was predominantly female (75.16%) with a mean age of 26.83 years (s.d. 11.26 years). Full details

of participant demographic and clinical information are detailed in Supplementary Information. Participants were recruited through databases held at each university (Edinburgh: the Patrick Wild Centre Participant Database; Nottingham: the Autism Research Team Database; and Dallas: The Autism Research Collaborative), partnerships with local autism charities and autistic organizations, and social media. Participants were remunerated (£30/US$40) for their time.

One hundred eight participants were allocated to the uninformed condition, and 216 participants were allocated to the informed condition. This allowed for a direct replication of the original study (where participants were informed about diagnostic status) with a sample size powered to detect smaller effects, while also permitting assessment of the effect of diagnostic information on task performance within the resources available for this study. The distribution of participants to these conditions was because of the funding constraints of this study. Specifically, we applied for funding to a scheme designed solely to fund replications and powered our study on that basis. In response to reviewer comments, the funder offered additional support to allow us to extend the study to include a smaller, uninformed diagnosis condition, and this resulted in the imbalanced groups.

**Inclusion criteria.** Autistic and non-autistic participants were required to be over 18 years, of any gender, to speak English to a native level and have normal/corrected normal sight and hearing. Consistent with the Crompton et al.[34] study, participants were ineligible if they had a diagnosis of social anxiety disorder or uncontrolled epilepsy. The autistic group included participants reporting a clinical diagnosis of autism ($n = 144$) and those who self-identified as being autistic ($n = 40$) provided they exceeded a clinical threshold using validated measures (see below). We anticipated that the likelihood of including significant numbers of self-identifying autistic people in this study was low given the availability of databases of people with confirmed diagnoses at each site, but are included here as a possibility if recruitment challenges were encountered. Participants who self-identified as autistic completed the Ritvo Autism and Asperger's Diagnostic Scale (RAADS)–Revised[70] and were included in the study if their score was above 72, as recommended in the literature[71]. All participants completed the RAADS 14-item screen (RAADS-14)[72], and non-autistic participants were excluded from participating if their scores indicated high levels of autistic traits (score >14).

**Chain types.** This study involved comparing performance on information transfer and rapport scores between three conditions (chain types) to which participants were assigned upon enrolment. The chain types were autistic chains, where all participants were autistic, non-autistic chains, where none of the participants were autistic, and mixed chains, where half the participants were autistic and half of the participants were not. We attempted to ensure that groups were comparable on gender, age, educational level, linguistic ability and IQ. However, as it was not possible to match the groups on gender and ethnicity, these variables were included as additional control variables on post hoc analysis as planned. These variables did not affect the results (Supplementary Information).

**Randomization.** This study was non-randomized, and participants were assigned to either a non-autistic, autistic or a mixed autistic–non-autistic chain and to the informed or uninformed condition according to gender, age and order of recruitment. Assignment was also affected by participant availability to attend on a particular data collection day.

**Revealing diagnostic status.** This study examined information transfer in two conditions: (1) the informed condition, where participants were aware of the diagnostic status of the participants in their diffusion chain group, and (2) the uninformed condition, where participants were not informed of the diagnostic status of the other participants in their diffusion type. The researchers were aware of the diagnostic status of all participants, and thus data collection, scoring and analysis was not performed blind to the conditions of the experiment.

**Content type.** This study examined whether the efficacy of information transfer differs for factual and fictional information. Content type was counterbalanced across the chains, with half of all chains first completing a fictional task and half first completing the factual task.

**Timeline.** Participants were recruited between August 2022 and October 2023.

## Procedure

**The experimental diffusion chain tasks.** This study used a diffusion chain methodology—a controlled, experimental form of the game 'Telephone'—which has been effective in probing cultural learning between individuals in a social group[73,74]. In this method, an experimenter models a complex behaviour to the first person in the chain; in this case, telling the participant a short passage of information. The person then has a chance to replicate the behaviour alone (that is, rehearse retelling the information) before being paired with the next person in the chain and instructed to demonstrate the behaviour to them (that is, retelling the passage). After hearing the passage, the second participant can practise the behaviour and then must pass it on to the next individual, and so on. Before commencing a diffusion chain, we ensured that consecutive participants in the chain did not know one another.

In practice, this meant that the researcher played a video to the first participant (A) in which a non-autistic man read the passage aloud. The researcher then left the room, and a second participant (B) entered. Participant A then recounted the story to participant B. Participant A then left the room, and a third participant (C) entered. Participant B then recounted the story to participant C, and so on, to the sixth participant (F). The sixth participant recounted the story aloud, alone. Participants waited in separate rooms for their turn, to avoid contamination during the information sharing. For mixed chains, half began with an autistic participant and half began with a non-autistic participant before alternating between autistic and non-autistic participants. All diffusion chains were video recorded for scoring purposes.

In each diffusion chain, six participants completed two separate diffusion chain tasks: a fictional and factual task. In each chain, six participants completed one task in full—for instance, passing a fictional story through all six people in the chain—and then the same chain in the same participant order completed the second task (in this case, factual information transfer). The order of fictional and factual task administration (within-group factor) was counterbalanced across chain types so that familiarity effects (from interacting with the same person twice) were distributed evenly between fictional and factual task conditions.

The fictional condition task was a short story that was surreal and difficult to predict. The factual condition task involved a short passage describing facts of an obscure scientific nature. The passages for both tasks feature 30 individual details, allowing the task to be scored out of a maximum of 30 for each participant. Both tasks had comparable Flesch–Kincaid grade level and Flesch reading ease scores, featured a comparable number of words and had comparable word and sentence lengths[75,76]. Both were designed to be completely novel to participants, difficult to predict and not involve any inherently social features.

A participant's final score corresponded to the number of details they recalled when recounting the passage to the next person in the chain, out of a maximum of 30. A higher score indicated a greater amount of information shared. Two researchers independently coded 50% of the videos. Additionally, they each second-coded 5% of randomly selected video material assigned to the other researcher, giving a 10% overlap of videos that were double coded. Inter-rater reliability was calculated using a single rating absolute-agreement two-way mixed-effects model as per ref. [77] and was very high for both

tasks (factual intraclass correlation coefficient 0.986 ($P < 0.0001$), 95% CI 0.975 to 0.992; fictional intraclass correlation coefficient 0.978 ($P < 0.0001$), 95% CI 0.961 to 0.987). Participants within each of the chains were ordered in ascending age to minimize a possible effect of age-related memory decline. Chains were also organized to minimize frequent switches of gender to avoid a possible effect on information transfer and rapport. Participants in the informed condition ($n = 216$) were told whether they were in an autistic, non-autistic or mixed chain. Participants in the uninformed condition ($n = 108$) were not informed of chain types. Participants did not meet before the study started and were isolated in separate rooms throughout the study, except when participating in the diffusion chains.

**The experimental rapport measure.** Participants were asked to rate the rapport they experienced while completing the diffusion chain tasks. Participants completed these rapport measures twice: once for the interaction when they were the 'listener' (that is, when they were listening to another participant recount the passage to them) and once for the interaction when they were the 'speaker' (that is, when they were recounting the passage to another participant). The first participant in each chain only rated rapport as a speaker, and the last participant in the chain only rated rapport as a listener. The rapport measure used was taken from the original study by Crompton et al.[34,40,41] and involved participants answering using a slider on a scale from 0 to 100: (1) how much did you enjoy the interaction? (2) how easy was the interaction? (3) how successful was the interaction? (4) how friendly was the interaction? (5) how awkward was the interaction? (reverse scored). The full measure can be found on the Open Science Framework website alongside the data for this study.

Studies of rapport in dyadic interactions typically use self-rated questionnaires[78]. While self-rated rapport may be subject to response biases (for example, if autistic people underestimate their rapport owing to negative self-perception of social skills or a history of difficult interactions with others, or if non-autistic people overestimate their rapport[40]), we consider that it is nevertheless the optimal way to assess each participants' direct experience of the interaction. Specifically, self-rated rapport was selected over observer-rated rapport, as most methods developed for measuring observer-rated rapport do not accommodate neurodiverse interactional experiences. External rapport measures can be biased by a neuro-normative lens: normative external indicators of rapport are less likely to be observed between autistic pairs[40], and this may be undetected or misinterpreted by observers. This means that, even if autistic pairs are experiencing high rapport, external observers are likely to rate them as having low rapport. For example, when independent observers rate videos of autistic people, they rate them as being more awkward and less approachable[14], both of which are key factors in building rapport. These biases are robust, are developed very rapidly and do not change with increased exposure[14]. Importantly, independent autistic observers have a similar tendency to non-autistic judges to evaluate autistic adults less favourably than non-autistic adults in videos[16], and so this bias cannot be overcome by simply recruiting both autistic and non-autistic independent judges. Similarly, emotion recognition is subject to strong neuro-normative biases – autistic people have different facial expressions to non-autistic people, and autistic people (and thus, emotion-recognition software based on non-autistic norms) are poor at identifying these emotions[28], which is related to their perceiving them unfavourably. Normative biases are also replicated within intelligent learning algorithms[79], and thus automated tools based in machine learning are similarly problematic to use in this context. Additionally, since we examined whether rapport varies depending on social context (single or mixed dyad), rather than as a main effect of diagnosis (autistic or non-autistic), any influence of response bias associated with autism was well managed by the study design. For these reasons, self-rated rapport was used in this study.

**Standardized tasks.** To characterize the IQ of the sample and match across groups, participants completed the Wechsler Abbreviated Scale of Intelligence II (WASI-II) two-subtest version[80]. Autistic participants had a mean IQ of 118.14 (s.d. 15.50), and non-autistic participants had a mean IQ of 111.46 (s.d. 12.95). A breakdown of group-level IQ scores by the chain condition (autistic, non-autistic and mixed) is available in Supplementary Information. All participants completed the RAADS-14 (ref. 72) to characterize the sample. Autistic participants had a mean score of 33.39 (s.d. 18.59), and non-autistic participants had a mean score of 5.15 (s.d. 4.17). A breakdown of RAADS scores and age of diagnosis by chain conditions that included autistic (autistic or mixed) is available in Supplementary Information. Additionally, participants who self-identified as autistic completed the RAADS-Revised[70] to ensure they scored above the threshold of 72, as recommended in the literature[72].

**Protocol.** Participants completed the tasks in the order shown in Table 1.

### Sampling plan

**Expected effect sizes.** There are few scientific comparisons between autistic, non-autistic and mixed social groups. The analysis in the original study gave a partial $\eta^2$ effect size of 0.45 for chain type, 0.83 for position and 0.08 for the interaction of chain type and position[34], though there are insufficient similar studies to know if these are reliable effect sizes. We therefore proposed being conservative in our effect size estimates given the paucity of data, especially in high-powered studies. In the proposed study, we suggested a sample size that is based on a revised data analysis and powered to detect medium effects in the data.

**Power analysis.** A mixed design for chains with both between and within factors appeared more appropriate than a between-subjects design. This increased the power of the study. We modified the linear model so that each chain (rather than participant) was treated as an independent observation and the proportion of recalled information from a participant in a chain was considered as a repeated measure, allowing for dependencies between participants within a chain. Applying a corresponding linear model with repeated measurements to the original data by Crompton et al.[34] suggested larger effect sizes for the main effects and interaction (chain type partial $\eta^2 = 0.52$, position in chain 0.87 and interaction 0.20). For equivalent analyses of rapport scores, we found a partial $\eta^2$ of 0.19 for the interaction.

Since the main effects were strong, the smallest meaningful interaction effect between chain type and position would be a medium effect of $\eta^2 = 0.06$ (partial Cohen's $f = 0.25$). To establish a correlation coefficient for the within-factor position, we had to introduce assumptions about the correlation matrix. For compound symmetry, we estimated $\rho = 0.502$ using a general least square fit (function gls() in R package nlme). Fitting an auto-regressive AR(1) correlation matrix to the data increased the coefficient to $\phi = 0.767$, but this fit was not significantly better than the compound symmetry fit. Assuming a statistical significance level of $P = 0.05$, a medium effect size of $\eta^2 = 0.06$, a lower correlation of $r = 0.4$ and a correction for non-sphericity of $\varepsilon = 0.7$ (Greenhouse–Geisser), then a power analysis[59] for a within–between interaction in an analysis of variance (ANOVA) with repeated measures 54 chains with 6 positions (participants) suggested a total of 324 participants to reach 95% power (see also Fig. 1).

A priori power analyses for linear mixed-effect models are notoriously difficult to conduct and require simulation studies[81,82]. The simulation-based power analysis required fitting a linear mixed model to the existing data by Crompton et al.[34], with eight participants in each of three chains for each condition ($N = 72$). Since this dataset has the minimum number of chains per condition, we could only fit a mixed model with random intercepts. If 'order' was added as a further random effect, then the model failed to converge and we could no longer run a simulation-based power analysis.

We calculated the exact power for the interaction effect of a mixed-effect linear model with a random intercept for each chain using the R packages lme4 (ref. [49]) and simr[83]. On the basis of the estimated coefficients of the mixed-effect model analysis on the original data (omitting the data for chain positions 7 and 8), Monte Carlo simulations gave power estimates for different numbers of chains. The simulation results were conservative because they are based on the estimated coefficients reduced by one-third. The simulations suggested more than 45 chains to test the fixed effect of interaction chain type by position with 95% power (Fig. 2).

Further details of the power analysis and simulations in R can be found in the R file PowerAnalysis alongside published data and code. The large sample of 324 participants was the maximum feasible under current funding constraints and provided sufficient power (>95%) to test the hypotheses and to explore undirected and two-way interaction effects on post hoc analyses. For the main hypotheses, we also computed BFs for normally distributed differences between means (R package BayesFactor[50]). Unlike Neyman–Pearson statistical inference, BFs accumulate evidence with increasing sample size and inform about the likelihood of the alternative relative to the null hypothesis given the evidence[84,85]. We also conducted comprehensive model comparisons using information criteria to establish the most parsimonious (mixed-effect/nonlinear) model[86].

Our sample (311 participants in 54 chains) was considerably larger than the original sample (72 participants in 9 chains) in ref. [34]. There are several benefits of this. First, we were powered (>95%) to detect and replicate results for reduced effect sizes. Second, this sample size provided us the opportunity to fit the maximal model[87] and/or to identify the most parsimonious model[87,88]—rather than relying on the intercept-only model used in the simulation. Third, it enabled us to examine further undirected effects, namely, comparing across informed and uninformed conditions, sites and content type with effect sizes that are likely to be smaller. Finally, it allowed us to account for potential data loss or outliers. The sample size was therefore substantially larger than the original study[34], and far exceeded the sample size of previous studies reporting similar group differences[28]. The increased sample and therefore number of chains was further justified because variability within chains could be investigated using not only linear, but also nonlinear and dynamic models (see post hoc analyses included in Supplementary Information).

**Adjusted significance levels.** A standard $P < 0.05$ threshold was used to determine statistical significance for testing in standard linear analyses. False discovery rate correction[89] was used to control for inflated alpha levels due to multiple comparisons, where applicable.

**Bayesian analysis.** Bayesian testing and modelling were carried out in R (for example, packages bayesfactor[50] and brms[90]). As there were no comparable previous studies to guide the exact specification of priors in these analyses, default priors were used for parameters and hypotheses. Sequential analysis, sensitivity analysis and predictive checks were carried out to ensure that default priors provide robust and accurate results[85]. Owing to the nature of the diffusion-chain task, observations inside a chain were not only dependent but were likely to follow a nonlinear decay function. On post hoc analyses, it was therefore planned to fit different decay functions and dependencies to further improve model fit. There were various options to establish nonlinear models to test between different chain conditions ranging from generalized linear mixed-effect models (R packages lme4 (ref. [49]) and nlme[91]), social reinforcement learning[92,93], auto-regressive models (R package fpp2 (ref. [94])) to stochastic (dyadic) process models (R packages brms[90] and rstan[76]) that can capture the transfer of information from one (autistic or non-autistic) person to the next (autistic or non-autistic) person inside each chain. We explored these analysis options to identify critical parameters and perform appropriate post hoc statistical testing.

**Data exclusion.** Outliers were identified as values ±2.5 s.d. from the mean in performance and rapport variables. These values were either (1) be left in if the distribution of the data still meets the assumption of the test (for example, adequate normality) or (2) excluded to make data meet the assumption of the test. In addition, (3) videos and notes were checked to see whether there is an explanation for outlying performance, and data were excluded if the underlying observation is non-representative (for example, task interrupted by an external event or a violation of the protocol). This was noted and discussed in reporting.

Using these criteria, no data were excluded from the analysis in the original study, and thus we hoped no data would need to be excluded from this replication. We made every effort to keep outliers in the dataset and select analyses that are robust to this, for example, by using type III sums of squares ANOVA, which are more tolerant to minor violations to assumptions of normality. However, three research days produced data of insufficient quality or quantity. Thus, the data from these days were not included, and the diffusion chains were re-run with new participants. Our reasons for excluding these data were (1) 1 day only had four participants attend, so there were not sufficient data to include, (2) one chain included a participant in position one who recalled a very low level of information, below the outlier threshold of ±2.5 s.d. from the mean outlined in our sampling plan, and (3) we had an unequal balance of missing data from five-person chains across the three conditions (autistic, non-autistic or mixed). To ensure that missing data were balanced across the three conditions, an additional autistic chain was re-run with six participants.

## Analysis plan

**Data.** Our analyses were based on two data sources: participant task performance data and self-rated rapport data. Additionally, demographic data was directly self-reported using a Qualtrics online form and downloaded into a .csv file. Next, we outline the proposed analysis pipeline for participant task performance data and self-rated rapport data, including preprocessing steps and planned analyses.

**Participant task performance data.** Participant interactions were videotaped, transcribed and then scored according to the number of details out of 30 transferred to the subsequent participant using the task scoring tool. Participant scores were then stored in a .csv data file and imported into R for analysis.

**Self-rated rapport data.** Participants self-rated their rapport experience out of 100 across five domains by using the rapport tool available alongside published data. These scores were entered into a datafile by a researcher and imported to R for analysis. The internal structure of the rapport construct was assessed using Cronbach's alpha, and analyses used a singular construct of rapport if the internal consistency was over an acceptable threshold (0.7), as recommended in the literature[71]. If any item caused the alpha to be below 0.7 it was dropped. From all items with an internal consistency of 0.7 or above, a mean was calculated, referred to as the 'rapport rating'. If items did not have high internal consistency, they were used in the following analysis individually, with false discovery rate corrections to avoid increased type 1 error risk.

**Analyses.** Hypotheses 1: (1a) participants in autistic and non-autistic chains will share significantly more details than in mixed chains; (1b) information will deteriorate between participant position (1–6) in each chain; and (1c) this should occur significantly earlier in mixed chains compared with autistic and non-autistic chains. Autistic and non-autistic chains will not differ in their deterioration rate.

We first examined the data, calculating descriptives (for example, mean and s.d.) and visualizations for the dependent variable (performance or rapport score) in each group (autistic, non-autistic or mixed) and checked for outliers. We then checked whether the five assumptions of linear regression are met for this model: (1) that the data were linear

(checked by inspecting a residuals versus fitted plot), (2) that residuals variance was homogeneous (checked by examining a spread-location plot), (3) that residuals were normally distributed (checked by examining a QQ plot), (4) that there was independence of residual error terms (checked by examining a scatter plot of residuals versus fits) and (5) collinearity (checked by residual plots and model fits). Next, we performed a linear regression on chains in mixed design, with the dependent variable of participant score and independent variables of chain type (between autistic, non-autistic and mixed) and chain position (within 1–6). This was compared with equivalent analyses on the original dataset. Additional post hoc analyses included nonlinear regression. The decay of information inside chains was unlikely to follow a linear function because loss of information may be larger at the start of the chain and smaller towards the end. Thus, a nonlinear regression model may significantly improve data fit. A nonlinear regression analysis essentially followed the same design as the linear regression: dependent variable of participant score and independent variables of chain condition (autistic, non-autistic and mixed) and position (1–6) in a mixed design.

Hypotheses 2: (2a) we predict significantly higher rapport scores in the autistic and non-autistic chains relative to the mixed chain; (2b) given that contact with autistic people is associated with more favourable impressions, less stigma and more inclusionary attitudes[20,71], we hypothesized that rapport will increase in the second relative to the first interaction between partners in mixed chains; and (2c) participants in the informed condition would have higher rapport scores than participants in the uninformed condition[46].

This analysis used data from both diffusion chain tasks completed by participants, whether fictional or factual. We examined and visualized the data and check the assumptions of the analysis method as described in hypothesis 1. We first performed a linear regression. This included the dependent variable of individual rapport score and predictor variables of chain type (autistic, non-autistic or mixed), interaction order (first or second) and diagnostic informing (informed or uninformed). We then investigated more complex linear models by adding one predictor at a time. In a comprehensive model comparison using information criteria, we compared the simplest established model with more complex models that include additional variables as well as two-way interactions to determine the most parsimonious linear model and its effects. We only interpreted the results of the most parsimonious model.

## Exploratory analyses
**Generalizability of findings.** In response to the reproducibility crisis[95], we addressed problems of generalizability[96] in the current project. Among the many recommendations to improve standards, we targeted sample size and power, alongside a more representative or ecologically valid design that addresses sampling of participants (three different sites), two different instructions (diagnostic informing) and stimulus material (content type). It was planned to emphasize variance estimates (instead of point estimates) and to fit alternative and more expansive statistical models (model comparison, mixed-effect models and nonlinear models). If similar effects of chain type and position were replicated under these broadened conditions and for different analyses, then we can rest assured that the effects hold true in the wider population.

**Generalizability of hypothesis 1.** To check whether there were differences in the data collected across the three sites, we established models that add predictor site (Edinburgh, Nottingham or Dallas), diagnostic information (informed or uninformed) and information content (fictitious or factual) as well as two-way interactions in a forwards selection. We performed model comparisons using information criteria to compare models and to determine the most parsimonious linear model and its effects.

**Generalizability of hypothesis 2.** To check whether there were differences in the data collected across the three sites, we established models

that add predictor site (Edinburgh, Nottingham or Dallas), participant role (whether they were speaking or listening) and information content (fictitious or factual), as well as two-way interactions in a forwards selection. We performed model comparisons using information criteria to compare models and to determine the most parsimonious linear model and its effects.

**Exploring potential effects of diagnostic informing and content type.** This study extended beyond ref. 34 by examining the potential impact of two manipulations: diagnostic informing (informed or uninformed) and content type (fictional or factual). Previous research has not clearly indicated whether diagnostic disclosure affects communication[16,46,97] nor is there sufficient evidence to suggest whether autistic or non-autistic people share fictional or factual information differently. Therefore, these hypotheses are exploratory and undirected. We examined and visualized the data and checked the assumptions of the analysis method, as described in hypotheses 1. We then performed a linear regression analysis using a simple linear model. This included the dependent variable of individual participant score and predictor variables of chain type (autistic, non-autistic or mixed) and chain position (1–6). We then added diagnostic informing (uninformed or informed), content type (fictional or factual) and site (Edinburgh, Dallas or Nottingham) as additional predictors. We created different linear models in a forwards selection by adding one predictor at a time. In a comprehensive model comparison using suitable information criteria, we compared the original model with more complex models that include more predictors and interactions to determine the most parsimonious linear model[80]. We only interpreted the output and effects of the most parsimonious model.

Additional post hoc analyses included nonlinear regression analyses. The decay of information inside all diffusion chains is unlikely to be linear; thus a nonlinear regression model may provide a better account of information loss across positions. The nonlinear regression followed the same design as the linear regression: dependent variable of individual participant score and predictors including chain type, chain position, diagnostic informing, information content and site. Similarly, testing utilized Bayesian statistics and model comparison to establish whether a nonlinear model outperforms a linear model (for example, package brms[90]).

**Exploring distortions in diffusion chain content.** In diffusion chains, the content of an original piece of information generally degrades over repeated social transmissions[73]. However, it is also possible that information becomes distorted, with participants adding detail not included in the original information (for example, "She turned left at the blue windmill" becomes "She turned left at the blue flowery windmill") or making an error in information transfer (for example, "She turned left at the blue windmill" becomes "She turned left at the blue castle"), which is then transferred to subsequent participants[92]. Thus, in addition to counting the number of correct details transmitted, we examined transcripts of interactions to quantify distortions occurring in information transfer. However, we were not able to arrive at a reliable coding scheme for deviations, as it was not possible to reach a moderate level of agreement among coders on what constituted a deviation. We attempted to code whether participants switched content (that is, removing one piece of content and replacing it with another) or introduced new content in addition to the existing content, but there were problems comparing substitutions between participants and defining what constituted a single 'unit' of deviation for scoring purposes was very complex. Though various coding schemes were posited, the application of these resulted in agreement consistently below 60%. This rate of agreement is too low to ensure data quality. As we could not meet a minimum threshold for data quality for coding this variable, this variable was not calculated or analysed[98].

We planned to examine and visualize the data and check the assumptions of the analysis method as described for hypothesis 1.

Our plan had been that, if the assumptions were met, then we would perform a linear regression analysis using a simple linear model and if not, we would apply Bayesian analysis methods. This would include the dependent variable of individual participant distortion score (that is, how many distortions each participant introduced), and predictor variables of chain type (autistic, non-autistic or mixed) and chain position (1–6). Collectively, the analyses could therefore determine both whether information degrades differently across chain types and whether new incorrect information was generated to a greater degree in some chain types (and by some participants) compared with others.

## Protocol registration

The Stage 1 protocol for this Registered Report was accepted in principle on 23 August 2022. The protocol, as accepted by the journal, can be found at https://osf.io/us9c7/.

## Reporting summary

Further information on research design is available in the Nature Portfolio Reporting Summary linked to this article.

## Data availability

All data and materials are freely and openly available via the Open Science Framework at https://osf.io/us9c7/.

## Code availability

All analysis code in R is freely and openly available via the Open Science Framework at https://osf.io/us9c7/.

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

## Acknowledgements

This work was funded by a grant from the Templeton World Charity Foundation (TWCF0442) awarded to C.J.C. (principal investigator), N.J.S., D.R. and S.F.-W. (co-investigators). Following peer review, the foundation offered additional funding to allow us to include a condition in which participants were uninformed about the diagnostic status of their interaction partner, as described in the Methods section. This was in response to recommendations from peer reviewers. Otherwise, the funders had no role in study design, data collection and analysis, decision to publish or preparation of the paper.

## Author contributions

C.J.C.: conceptualization, funding acquisition, methodology, project administration, investigation, methodology, writing—original draft, writing—reviewing and editing, and supervision. S.J.F.: data collection, project administration and writing—reviewing and editing. C.E.H.W.: data collection, project administration, writing—reviewing and editing. M.D.: data collection, project administration and writing—reviewing and editing. T.N.E.: data curation, visualization and writing—reviewing and editing. N.J.S.: conceptualization, funding acquisition, methodology, project administration, investigation, methodology, writing—reviewing and editing, and supervision. D.R.: conceptualization, funding acquisition, methodology, project administration, investigation, methodology, writing—reviewing and editing, and supervision. M.L.: methodology, visualization, formal analysis, writing—original draft and writing—reviewing and editing. S.F.-W.: conceptualization, funding acquisition, methodology, methodology and writing—reviewing and editing

## Competing interests

The authors declare no competing interests.

## Additional information

**Correspondence and requests for materials** should be addressed to Catherine J. Crompton.

# Reporting Summary

## Statistics

For all statistical analyses, confirm that the following items are present in the figure legend, table legend, main text, or Methods section.

| n/a | Confirmed | |
|---|---|---|
| ☐ | ☒ | The exact sample size (*n*) for each experimental group/condition, given as a discrete number and unit of measurement |
| ☐ | ☒ | A statement on whether measurements were taken from distinct samples or whether the same sample was measured repeatedly |
| ☐ | ☒ | The statistical test(s) used AND whether they are one- or two-sided *Only common tests should be described solely by name; describe more complex techniques in the Methods section.* |
| ☐ | ☒ | A description of all covariates tested |
| ☐ | ☒ | A description of any assumptions or corrections, such as tests of normality and adjustment for multiple comparisons |
| ☐ | ☒ | A full description of the statistical parameters including central tendency (e.g. means) or other basic estimates (e.g. regression coefficient) AND variation (e.g. standard deviation) or associated estimates of uncertainty (e.g. confidence intervals) |
| ☐ | ☒ | For null hypothesis testing, the test statistic (e.g. *F*, *t*, *r*) with confidence intervals, effect sizes, degrees of freedom and *P* value noted *Give P values as exact values whenever suitable.* |
| ☐ | ☒ | For Bayesian analysis, information on the choice of priors and Markov chain Monte Carlo settings |
| ☐ | ☒ | For hierarchical and complex designs, identification of the appropriate level for tests and full reporting of outcomes |
| ☐ | ☒ | Estimates of effect sizes (e.g. Cohen's *d*, Pearson's *r*), indicating how they were calculated |

*Our web collection on statistics for biologists contains articles on many of the points above.*

## Software and code

Policy information about availability of computer code

| Data collection | No software was used during data collection |
|---|---|
| Data analysis | All code for analysis is freely and openly available on the Open Science Framework page for this study (https://osf.io/us9c7/) |

For manuscripts utilizing custom algorithms or software that are central to the research but not yet described in published literature, software must be made available to editors and reviewers. We strongly encourage code deposition in a community repository (e.g. GitHub). See the Nature Portfolio guidelines for submitting code & software for further information.

Analyses were conducted in R version 4.4.2, using packages lme4: Version 1.1.35.5; bayesfactor: Version 0.9.12.4.7; MuMIn: Version 1.48.4; r2glmm: Version 0.1.2; lmerTest: Version 3.1.3; simr: Version 1.0.7; brms: Version 2.22.0; nlme: Version 3.1.166; fpp2: Version 2.5; rstan: Version 2.32.6

## Data

Policy information about availability of data

All manuscripts must include a data availability statement. This statement should provide the following information, where applicable:
- Accession codes, unique identifiers, or web links for publicly available datasets
- A description of any restrictions on data availability
- For clinical datasets or third party data, please ensure that the statement adheres to our policy

All data are freely and openly available on the Open Science Framework page for this study (https://osf.io/us9c7/)

# Research involving human participants, their data, or biological material

Policy information about studies with <u>human participants or human data.</u> See also policy information about <u>sex, gender (identity/presentation), and sexual orientation</u> and <u>race, ethnicity and racism.</u>

| | |
|---|---|
| **Reporting on sex and gender** | We attempted to recruit a sample that was representative of a range of genders, and attempted to ensure that the autistic and non-autistic groups were comparable on gender.<br><br>Participants self-reported their gender on an initial demographic questionnaire. Of the 311 participants, 197 (63.34%) were women, 58 (18.65%) were men, 50 (16.07%) were non-binary, and 6 (1.94%) preferred not to disclose their gender or self-identified as another gender. Of the autistic participants specifically (n = 154), 79 (51.30%) were women, 23 (14.94%) were men, 47 (30.52%) were non-binary, and 5 (3.25%) preferred not to disclose or self-identified as another gender. Of the non-autistic participants specifically (n = 157), 118 (75.16%) were women, 35 (22.29%) were men, 3 (1.91%) were non-binary, and 1 (0.64%) preferred not to disclose or self-identified as another gender. A breakdown of participant genders by the chain condition (autistic, non-autistic, and mixed) is available in the Supplement.<br><br>We do not anticipate any differences on performance based on gender. However, as the groups significantly differed on gender, gender was included as an additional control variable in post-hoc analysis, but did not affect the results. This is described in the Supplement. |
| **Reporting on race, ethnicity, or other socially relevant groupings** | We attempted to recruit a sample that was representative of a range of ethnicities, and attempted to ensure that the autistic and non-autistic groups were comparable on ethnicity.<br><br>Participants self-reported their ethnicity on an initial demographic questionnaire. These questionnaires used census-based frameworks for each of the countries where data were collected (USA, England, Scotland) and were then collapsed into higher-level categories to enhance comparability.<br><br>Of the 311 participants, 203 (65.27%) were White, 65 (20.90%) were Asian, 19 (6.11%) were Mixed or Multiple Ethnicities, 13 (4.18%) were Black, 5 (1.61%) were Hispanic, and 6 (1.93%) identified as 'Other'. Of the autistic participants specifically (n = 154), 116 (75.32%) were White, 14 (9.09%) were Asian, 11 (7.14%) were Mixed or Multiple Ethnicities, 8 (5.19%) were Black, 3 (1.95%) were Hispanic, and 2 (1.30%) identified as 'Other'. Of the non-autistic participants specifically (n = 157), 87 (55.41%) were White, 51 (32.48%) were Asian, 8 (5.10%) were Mixed or Multiple Ethnicities, 5 (3.18%) were Black, 2 (1.27%) were Hispanic, and 4 (2.55%) identified as 'Other'. A breakdown of participant ethnicities by the chain condition (autistic, non-autistic, and mixed) is available in the Supplement.<br><br>We do not anticipate any differences on performance based on ethnicity. However, as the groups significantly differed on ethnicity, ethnicity was included as an additional control variable in post-hoc analysis, but did not affect the results. This is described in the Supplement. |
| **Population characteristics** | We attempted to recruit a sample that was representative of a range of ages, and attempted to ensure that the autistic and non-autistic groups were comparable on age. Autistic participants had a mean age of 28.68 (SD = 11.18) and non-autistic participants had a mean age of 26.83 (SD = 11.26). A breakdown of participant age by the chain condition (autistic, non-autistic, and mixed) is available in the Supplement.<br><br>Participants provided details of the highest educational level achieved during their initial demographic questionnaire, and both groups were matched on educational level (see Supplement for full details).<br><br>Of the 311 participants, 5 (1.61%) had not completed high school, 54 (17.36%) had completed high school, 18 (5.79%) completed community college or vocational qualifications, 158 (50.80%) had a partially completed or completed undergraduate degree, and 84 (27.00%) had a partially completed or completed postgraduate degree. Of the autistic participants specifically (n = 154), 4 (2.60%) had not completed high school, 25 (16.23%) had completed high school, 10 (6.49%) completed community college or vocational qualifications, 74 (48.05%) had a partially completed or completed undergraduate degree, and 41 (26.63%) had a partially completed or completed postgraduate degree. Of the non-autistic participants specifically (n = 157), 1 (0.64%) had not completed high school, 29 (18.47%) had completed high school, 8 (5.10%) completed community college or vocational qualifications, 76 (48.41%) had a partially completed or completed undergraduate degree, and 43 (27.39%) had a partially completed or completed postgraduate degree. Details of the educational levels for autistic participants and non-autistic participants presented separately are available in the Supplement, along with a breakdown of participant education level by the chain condition (autistic, non-autistic, and mixed).<br><br>Participant IQ was measured using the Wechsler Abbreviated Scale of Intelligence-II. Autistic participants had a mean IQ of 118.14 (SD = 15.50), and non-autistic participants had a mean IQ of 111.46 (SD = 12.95). A breakdown of group-level IQ scores by the chain condition (autistic, non-autistic, and mixed) is available in the Supplement.<br><br>The autistic group included participants reporting a clinical diagnosis of autism (n = 144) and those who self-identify as being autistic (n = 40). Autistic participants mean age of diagnosis (or self-diagnosis) was 23.72 years (SD = 12.68). Participants who self-identifed as autistic completed the Ritvo Autism and Asperger's Diagnostic Scale-Revised (RAADS-R)48 and were included if their score was above 72, as recommended in the literature. All participants completed the Ritvo Autism and Asperger's Diagnostic Scale 14-item Screen (RAADS-14). Autistic participants had a mean score of 33.39 (SD = 18.59); non-autistic participants had a mean score of 5.15 (SD = 4.17). A breakdown of RAADS scores and age-of diagnosis by chain conditions that included autistic (autistic, mixed)) is available in the Supplement. |

| Recruitment | 324 adult participants were recruited across three sites (University of Edinburgh, University of Nottingham, and the University of Texas at Dallas) and 311 (154 autistic, 157 non-autistic) attended research days. Participants were recruited through databases held at each University (Edinburgh: the Patrick Wild Centre Participant Database; Nottingham: the Autism Research Team Database; Dallas: The Autism Research Collaborative), partnerships with local autism charities and autistic organisations, and social media. Due to the recruitment being largely via Universities, it is likely that our sample is relatively highly educated, which may impact results. This point is included in the Discussion. |
|---|---|
| Ethics oversight | This study was carried out in accordance with the British Psychological Society's Code on Human Research Ethics and the American Psychological Association's Ethical Principles of Psychologists and Code of Conduct. Experimental procedures were reviewed and approved by the University of Edinburgh's Medical Research Ethics Committee, University of Nottingham, School of Psychology Ethics Committee, and the University of Texas at Dallas's Institutional Review Board. All participants provided written informed consent before participating and were remunerated for their time (£30/$40). |

Note that full information on the approval of the study protocol must also be provided in the manuscript.

# Field-specific reporting

Please select the one below that is the best fit for your research. If you are not sure, read the appropriate sections before making your selection.

☐ Life sciences   ☒ Behavioural & social sciences   ☐ Ecological, evolutionary & environmental sciences

For a reference copy of the document with all sections, see nature.com/documents/nr-reporting-summary-flat.pdf

# Behavioural & social sciences study design

All studies must disclose on these points even when the disclosure is negative.

| Study description | This study used a mixed experimental design incorporating both between and within-groups factors. Between-groups factors included chain type (autistic, non-autistic, mixed) and diagnostic-informing (informed, uninformed), with task type (fictional, factual) as a within-group factor.

Diffusion chain tasks and the Rapport task generated quantitative data (see Data Collection below for full details). |
|---|---|
| Research sample | Autistic and non-autistic participants were older than 18 years, of any gender, spoke English equivalent to a native level, and had normal/corrected normal sight and hearing. Participants were ineligible if they had a diagnosis of Social Anxiety Disorder or uncontrolled epilepsy. All participants provided written informed consent before participating and were remunerated for their time .

All participants completed the Ritvo Autism and Asperger's Diagnostic Scale 14-item Screen (RAADS-14), and non-autistic participants were excluded from participating if their scores indicated high levels of autistic traits (score > 14).

We aimed to recruit a diverse, representative sample including a range of genders, ages, ethnicities, and intellectual abilities. Our sample does have a fairly high level of education and IQ, and so results should be interpreted with this in mind. |
| Sampling strategy | Participants were recruited using convenience sampling.

Sample size was determined using an a-priori power analysis outlined in our Stage 1 submission, and is detailed below.

Expected effect sizes: There are few scientific comparisons between autistic, non-autistic, and mixed social groups. The analysis in the original study gave a partial h2 effect size of 0.45 for chain type, 0.83 for position, and 0.08 for the interaction of chain type and position, though there are insufficient similar studies to know if these are reliable effect sizes. We, therefore, propose being conservative in our effect size estimates given the paucity of data, especially in high-powered studies. In the proposed study we suggest a sample size that is based on a revised data analysis and powered to detect medium effects in the data.

Power analysis: A mixed design for chains with between and within factors appears more appropriate than a between-subjects design. This increases the power of the study. We modified the linear model so that each chain (rather than participant) is treated as an independent observation and proportion of recalled information from a participant in a chain is considered as a repeated measure, allowing for dependencies between participants within a chain. Applying a corresponding linear model with repeated measurements to the original data by Crompton et al. (2020) suggests larger effect sizes for the main effects and interaction (chain type partial h2 =0.52, position in chain 0.87, interaction 0.20). For equivalent analyses of rapport scores we found a partial h2 of 0.19 for the interaction.)

Since the main effects are strong, the smallest meaningful interaction effect between chain type and position would be a medium effect of h2 =0.06 (partial Cohen's f =0.25). In order to establish a correlation coefficient for the within-factor position we have to introduce assumptions about the correlation matrix. For compound symmetry we estimated r =0.502 using a general least square fit (function gls() in R-package nlme. (Fitting an auto-regressive AR(1) correlation matrix to the data increased the coefficient to φ =0.767 but this fit was not significantly better than the compound symmetry fit.)  Assuming a statistical significance level of p=0.05, a medium effect size of h2 =0.06, a lower correlation of r=0.4 and a correction for non-sphericity of e =0.7 (Greenhouse-Geisser) then a power analysis59 for a within-between interaction in an ANOVA with repeated measures 54 chains with 6 positions (participants) |

suggests a total of 324 participants to reach 95% power (see also Figure 1 at https://osf.io/us9c7/ ).

A priori power analyses for linear mixed-effect models are notoriously difficult to conduct and require simulation studies. A simulation-based power analysis requires fitting a linear mixed model to the existing data by Crompton et al., with 8 participants in each of 3 chains for each condition (N=72). Since this data set has the minimum number of chains per condition we can only fit a mixed model with random intercepts. If 'position' is added as a further random effect then the model fails to converge and we can no longer run a simulation-based power analysis.

We calculated the exact power for the interaction effect of a mixed-effect linear model with a random intercept for each chain using the R-packages lme4 and simr. Based on the estimated coefficients of the mixed-effect model analysis on the original data (omitting the data for chain position 7 and 8) Monte-Carlo simulations gave power estimates for different numbers of chains. The simulation results are conservative because they are based on the estimated coefficients reduced by one third. The simulations suggest more than 45 chains to test the fixed effect of interaction chain type by position with 95% power (see Figure 2 at https://osf.io/us9c7/).

Further details of the power analysis and simulations in R can be found in the R-file PowerAnalysis at https://osf.io/us9c7/

The large sample of 324 participants is the maximum feasible under current funding constraints and should provide sufficient power (>95%) to test the hypotheses and to explore undirected and two-way interaction effects in post-hoc analyses. For the main hypotheses we will also compute Bayes factors for normally distributed differences between means (R-package BayesFactor). Unlike Neyman-Pearson statistical inference Bayes factors accumulate evidence with increasing sample size and inform about the likelihood of the alternative relative to the null hypothesis given the evidence. We will also conduct comprehensive model comparisons using information criteria to establish the most parsimonious (mixed-effect/non-linear) model.

Our proposed sample (324 participants in 54 chains) is considerably larger than the original sample (72 participants in 9 chains) in Crompton et al. There are several benefits of this. First, we are powered (>95%) to detect and replicate results for reduced effect sizes. Second, this sample size should also give us the opportunity to fit the maximal model and/or to identify the most parsimonious model – rather than relying on the intercept-only model used in the simulation. Third, it enables us to examine further undirected effects, namely comparing across informed and uninformed conditions, sites, and content type with effect sizes that are likely to be smaller. Finally, it allows us to account for potential data loss or outliers. The sample size is therefore substantially larger than the original study, and far exceeds the sample size of previous studies reporting similar group differences. The increased sample and therefore number of chains is further justified because variability within chains can be investigated using not only linear but also non-linear and dynamic models (see post-hoc analyses).

**Data collection**

Chain types: This study involved comparing performance on information transfer and rapport scores between three groups(chain types) to which participants were assigned upon enrolment. The chain types were: autistic chains, where all participants are autistic; non-autistic chains, where all participants are non-autistic, and mixed chains, where half the participants are autistic and half the participants are non-autistic.

Revealing diagnostic status: This study examined information transfer in two conditions; (1) the informed condition, where participants were aware of the diagnostic status of the participants in their diffusion chain group and (2) the uninformed condition, where participants were not informed of the diagnostic status of the other participants in their diffusion type. 105 participants were allocated to the uninformed condition, and 206 participants were allocated to the informed condition. This facilitated a direct replication of the original study (where participants were informed about diagnostic status) with a sample size powered to detect smaller effects, while also permitting assessment of the effect of diagnostic-information on information transfer within the resources available for this study. The distribution of participants to these conditions was because of the funding constraints of this study. Specifically, we applied for funding to a scheme designed solely to fund replications and powered our study on that basis. In response to reviewer comments the funder offered additional support to allow us to extend the study to include a smaller, uninformed diagnosis condition, and this resulted in the imbalanced groups.

Content type: This study examined whether the efficacy of information transfer differs for factual and fictional information. Content type was counterbalanced across the chains, with half of all chains first completing a fictional task, and half first completing the factual task.

Procedure
The experimental diffusion chain tasks: This study used a diffusion chain methodology - a controlled, experimental form of the game "Telephone" - which has been effective in probing cultural learning between individuals in a social group52,53. In this method, an experimenter models a complex behaviour to the first person in the chain; in this case, verbally telling the participant a short passage of text. The person then has a chance to rehearse re-telling the information alone before being paired with the next person in the chain and instructed tell them the passage. After hearing the passage, the second participant can practice the behaviour and then must pass it on to the next individual, and so on. Before commencing a diffusion chain, we ensured that consecutive participants in the chain did not know one another.

In practice, this meant that the researcher played a video to the first participant (A) where a non-autistic man read the passage aloud. The researcher then left the room and a second participant (B) entered. Participant A then recounted the story to Participant B. A then left the room, and a third participant (C) entered. Participant B then recounted the story to Participant C and so on, to the sixth participant (F). The sixth participant recounted the story aloud, alone. Participants waited in separate rooms for their turn, to avoid contamination during the information sharing. For mixed chains, half began with an autistic participant and half began with a non-autistic participant before alternating between autistic and non-autistic participants. All diffusion chains were video recorded for scoring purposes.

In each diffusion chain six participants completed two separate diffusion chain tasks: a fictional and factual task. In each chain six participants completed one task in full – for instance passing a fictional story through all six people in the chain - and then the same chain in the same participant order completed the second task. The order of fictional and factual task administration (within group factor) was counterbalanced across chain types so that familiarity effects (from interacting with the same person twice) were

distributed evenly between fictional and factual task conditions.

The fictional condition task was a short story which was surreal and difficult to predict. The factual condition task involved a short passage describing facts of an obscure scientific nature. The passages for both tasks feature 30 individual details, allowing the task to be scored out of a maximum of 30 for each participant. Both tasks had comparable Flesch-Kincaid Grade Level and Flesch Reading Ease scores, featured a comparable number of words, and had comparable word and sentence lengths[54,55]. Both were designed to be completely novel to participants, difficult to predict and not involve any inherently social features.

A participant's final score corresponded to the number of details they recalled when recounting the passage to the next person in the chain, out of a maximum of 30. A higher score indicated a greater amount of information shared. Both passages and their scoring schemes are available on the study OSF website (https://osf.io/us9c7/), though were embargoed while data collection was underway to ensure that participants were unfamiliar with the content of the passages when they participated in the study. Two researchers independently coded 50% of the videos. Additionally, they each second coded 5% of randomly selected video material assigned to the other researcher, giving a 10% overlap of videos that were double coded. Inter-rater reliability was calculated using a Single Rating Absolute-Agreement 2-way Mixed-effects model as per[56], and was very high for both tasks (factual ICC 0.986 ($p<.0001$), 95% CI [0.975, 0.992]; fictional ICC .978 ($p<.0001$), 95% CI [0.961, 0.987]).

Participants within each of the chains were ordered in ascending age, to minimise a possible effect of age-related memory decline. Chains were also organised to minimise frequent switches of gender in order to avoid a possible effect on information transfer and rapport.

Participants in the informed condition were told whether they were in an autistic, non-autistic, or mixed chain. Participants in the uninformed condition were not informed about chain types. Participants did not meet before the study started and waited in separate rooms throughout the study, except when participating in the diffusion chains.

The experimental rapport measure: Participants were asked to rate the rapport they experienced while completing the diffusion chain tasks. Participants completed these rapport measures twice: once for the interaction when they were the 'listener' (i.e., when they were listening to another participant recount the passage to them) and once for the interaction when they were the 'speaker' (i.e., when they were recounting the passage to another participant). The first participant in each chain only rated rapport as a speaker, and the last participant in the chain only rated rapport as a listener.

The rapport measure used was taken from the original Crompton et al. study[34,40,41] and involved participants answering using a slider on a scale from 0 to 100 (1) how much did you enjoy the interaction? (2) how easy was the interaction? (3) how successful was the interaction? (4) how friendly was the interaction? (5) how awkward was the interaction? (reverse scored). The full measure can be found on the OSF website (https://osf.io/us9c7/).

Studies of rapport in dyadic interactions typically use self-rated questionnaires[57]. While self-rated rapport may be subject to response biases (for example, if autistic people underestimate their rapport due to negative self-perception of social skills or a history of difficult interactions with others; or if non-autistic people overestimate their rapport[40], we consider that it is nevertheless the optimal way to assess each participants' direct experience of the interaction.

Specifically, self-rated rapport was selected over observer-rated rapport, as most methods developed for measuring observer-rated rapport do not accommodate neurodiverse interactional experiences. External rapport measures can be biased by a neuro-normative lens: normative external indicators of rapport are less likely to be observed between autistic pairs[40], and this may be undetected or misinterpreted by observers. This means that even if autistic pairs are experiencing high rapport, external observers are likely to rate them as having low rapport.

For example, when independent observers rate videos of autistic people, they rate them as being more awkward and less approachable[14], both of which are key factors in building rapport. These biases are robust, are developed very rapidly, and do not change with increased exposure[14]. Importantly, autistic independent observers have a similar tendency to non-autistic judges to evaluate autistic adults less favourably than non-autistic adults in videos[16], and so this bias cannot be overcome by simply recruiting both autistic and non-autistic independent judges. Similarly, emotion recognition is subject to strong neuro-normative biases – autistic people have different facial expressions to non-autistic people, and non-autistic people (and thus, emotion recognition software based on non-autistic norms) are poor at identifying these emotions[28], which is related to their perceiving them unfavourably. Normative biases have also been detected within intelligent learning algorithms[58], and thus automated tools based on machine learning are similarly problematic in this context.

Additionally, since we examined whether rapport varies depending on social context (single or mixed dyad), rather than as a main effect of diagnosis (autistic or non-autistic), any influence of response bias associated with autism was well-managed by the study design. For these reasons, self-rated rapport was used in this study.

Standardised tasks: To characterise the IQ of the sample and match across groups, participants completed the Wechsler Abbreviated Scale of Intelligence II (WASI-II) two-subtest version[59]. All participants completed the Ritvo Autism and Asperger's Diagnostic Scale 14-item Screen (RAADS-14)[50] to characterise the sample; additionally, participants who self-identify as autistic completed the Ritvo Autism and Asperger's Diagnostic Scale-Revised (RAADS-R)[48].

Protocol: Participants completed the tasks in the following order.

Online – in advance of in person participation
- Information sheet and consent form
- Demographic questionnaire
- Ritvo Autism and Asperger's Diagnostic Scale (Ritvo et al., 2011) / Ritvo Autism and Asperger's Diagnostic Scale 14-item Screen (RAADS-14; Eriksson et al., 2013)
-
In person participation
- WASI – II (Wechsler, 2011)

April 2023

- Diffusion Chain Task 1
- Rapport Measure 1
- Diffusion Chain Task 2
- Rapport Measure 2

The Stage 1 protocol for this Registered Report was accepted in principle on 23rd August 2022. The protocol, as accepted by the journal, can be found at https://osf.io/us9c7/

Researcher knowledge: Researchers were aware of the diagnostic status of all participants, and thus data collection, scoring, and analysis was not performed blind to the conditions of the experiment.

| | |
|---|---|
| Timing | Participants were recruited between August 2022 and October 2023. |
| Data exclusions | Three research days produced data of insufficient quality or quantity. Thus, the research from these days were not included, and the diffusion chains re-run with new participants. Our reasons for excluding these data were (1) one day only had four participants attend, so there was not sufficient data to include (2) one chain included a participant in position one who recalled a very low level of information, below the outlier threshold of ±2.5 standard deviations from the mean outlined in our Sampling Plan, and (3) we had an unequal balance of missing data from five-person chains across the three conditions (autistic, non-autistic, mixed); in order to ensure that missing data was balanced across the three conditions, an additional autistic chain was re-run with six participants. |
| Non-participation | We aimed to recruit 324 participants (18 chains consisting of 6 people in each of the autistic, non-autistic, and mixed conditions). Due to a small number of participants (n =13) not attending, 13 chains contained only five participants (5 autistic, 4 non-autistic, 4 mixed). No reason was given by participants for their non-attendance.<br><br>Our power analysis suggests that our planned analyses are robust for up to 5.5% missing values; with our actual missing data of 4% below this threshold. |
| Randomization | This study used non-randomised samples, and participants were assigned to either a non-autistic, autistic, or a mixed autistic-non-autistic chain and to the informed or uninformed condition according to diagnostic status, gender, order of recruitment, and participant availability.<br><br>The order of fictional task first of factual task first was counterbalanced across the diffusion chains, and accounted for in the analysis by its inclusion as a predictor variable |

# Reporting for specific materials, systems and methods

We require information from authors about some types of materials, experimental systems and methods used in many studies. Here, indicate whether each material, system or method listed is relevant to your study. If you are not sure if a list item applies to your research, read the appropriate section before selecting a response.

## Materials & experimental systems

| n/a | Involved in the study |
|---|---|
| x | Antibodies |
| x | Eukaryotic cell lines |
| x | Palaeontology and archaeology ☐ |
| X | Animals and other organisms |
| X | Clinical data |
| X | Dual use research of concern |
| X | Plants |

## Methods

| n/a | Involved in the study |
|---|---|
| X | ChIP-seq |
| X | Flow cytometry |
| X | MRI-based neuroimaging |

## Antibodies

| Antibodies used | Describe all antibodies used in the study; as applicable, provide supplier name, catalog number, clone name, and lot number. |
|---|---|
| Validation | Describe the validation of each primary antibody for the species and application, noting any validation statements on the manufacturer's website, relevant citations, antibody profiles in online databases, or data provided in the manuscript. |

## Eukaryotic cell lines

Policy information about cell lines and Sex and Gender in Research

| Cell line source(s) | State the source of each cell line used and the sex of all primary cell lines and cells derived from human participants or vertebrate models. |
|---|---|
| Authentication | Describe the authentication procedures for each cell line used OR declare that none of the cell lines used were authenticated. |

| Mycoplasma contamination | *Confirm that all cell lines tested negative for mycoplasma contamination OR describe the results of the testing for mycoplasma contamination OR declare that the cell lines were not tested for mycoplasma contamination.* |
|---|---|
| Commonly misidentified lines<br>(See ICLAC register) | *Name any commonly misidentified cell lines used in the study and provide a rationale for their use.* |

# Palaeontology and Archaeology

| Specimen provenance | *Provide provenance information for specimens and describe permits that were obtained for the work (including the name of the issuing authority, the date of issue, and any identifying information). Permits should encompass collection and, where applicable, export.* |
|---|---|
| Specimen deposition | *Indicate where the specimens have been deposited to permit free access by other researchers.* |
| Dating methods | *If new dates are provided, describe how they were obtained (e.g. collection, storage, sample pretreatment and measurement), where they were obtained (i.e. lab name), the calibration program and the protocol for quality assurance OR state that no new dates are provided.* |

☐ Tick this box to confirm that the raw and calibrated dates are available in the paper or in Supplementary Information.

| Ethics oversight | *Identify the organization(s) that approved or provided guidance on the study protocol, OR state that no ethical approval or guidance was required and explain why not.* |
|---|---|

Note that full information on the approval of the study protocol must also be provided in the manuscript.

# Animals and other research organisms

Policy information about studies involving animals; ARRIVE guidelines recommended for reporting animal research, and Sex and Gender in Research

| Laboratory animals | *For laboratory animals, report species, strain and age OR state that the study did not involve laboratory animals.* |
|---|---|
| Wild animals | *Provide details on animals observed in or captured in the field; report species and age where possible. Describe how animals were caught and transported and what happened to captive animals after the study (if killed, explain why and describe method; if released, say where and when) OR state that the study did not involve wild animals.* |
| Reporting on sex | *Indicate if findings apply to only one sex; describe whether sex was considered in study design, methods used for assigning sex. Provide data disaggregated for sex where this information has been collected in the source data as appropriate; provide overall numbers in this Reporting Summary. Please state if this information has not been collected. Report sex-based analyses where performed, justify reasons for lack of sex-based analysis.* |
| Field-collected samples | *For laboratory work with field-collected samples, describe all relevant parameters such as housing, maintenance, temperature, photoperiod and end-of-experiment protocol OR state that the study did not involve samples collected from the field.* |
| Ethics oversight | *Identify the organization(s) that approved or provided guidance on the study protocol, OR state that no ethical approval or guidance was required and explain why not.* |

Note that full information on the approval of the study protocol must also be provided in the manuscript.

# Clinical data

Policy information about clinical studies

All manuscripts should comply with the ICMJE guidelines for publication of clinical research and a completed CONSORT checklist must be included with all submissions.

| Clinical trial registration | *Provide the trial registration number from ClinicalTrials.gov or an equivalent agency.* |
|---|---|
| Study protocol | *Note where the full trial protocol can be accessed OR if not available, explain why.* |
| Data collection | *Describe the settings and locales of data collection, noting the time periods of recruitment and data collection.* |
| Outcomes | *Describe how you pre-defined primary and secondary outcome measures and how you assessed these measures.* |

# Dual use research of concern

Policy information about dual use research of concern

## Hazards

Could the accidental, deliberate or reckless misuse of agents or technologies generated in the work, or the application of information presented in the manuscript, pose a threat to:

No | Yes

☐ Public health

☐ National security

☐ Crops and/or livestock

☐ Ecosystems

☐ Any other significant area

## Experiments of concern

Does the work involve any of these experiments of concern:

No | Yes

☐ Demonstrate how to render a vaccine ineffective

☐ Confer resistance to therapeutically useful antibiotics or antiviral agents

☐ Enhance the virulence of a pathogen or render a nonpathogen virulent

☐ Increase transmissibility of a pathogen

☐ Alter the host range of a pathogen

☐ Enable evasion of diagnostic/detection modalities

☐ Enable the weaponization of a biological agent or toxin

☐ Any other potentially harmful combination of experiments and agents

# Plants

| Seed stocks | Report on the source of all seed stocks or other plant material used. If applicable, state the seed stock centre and catalogue number. If plant specimens were collected from the field, describe the collection location, date and sampling procedures. |
| --- | --- |
| Novel plant genotypes | Describe the methods by which all novel plant genotypes were produced. This includes those generated by transgenic approaches, gene editing, chemical/radiation-based mutagenesis and hybridization. For transgenic lines, describe the transformation method, the number of independent lines analyzed and the generation upon which experiments were performed. For gene-edited lines, describe the editor used, the endogenous sequence targeted for editing, the targeting guide RNA sequence (if applicable) and how the editor was applied. |
| Authentication | Describe any authentication procedures for each seed stock used or novel genotype generated. Describe any experiments used to assess the effect of a mutation and, where applicable, how potential secondary effects (e.g. second site T-DNA insertions, mosaicism, off-target gene editing) were examined. |

# ChIP-seq

## Data deposition

☐ Confirm that both raw and final processed data have been deposited in a public database such as GEO.

☐ Confirm that you have deposited or provided access to graph files (e.g. BED files) for the called peaks.

| Data access links<br>May remain private before publication. | For "Initial submission" or "Revised version" documents, provide reviewer access links. For your "Final submission" document, provide a link to the deposited data. |
| --- | --- |
| Files in database submission | Provide a list of all files available in the database submission. |
| Genome browser session<br>(e.g. UCSC) | Provide a link to an anonymized genome browser session for "Initial submission" and "Revised version" documents only, to enable peer review. Write "no longer applicable" for "Final submission" documents. |

## Methodology

| Replicates | Describe the experimental replicates, specifying number, type and replicate agreement. |
| --- | --- |
| Sequencing depth | Describe the sequencing depth for each experiment, providing the total number of reads, uniquely mapped reads, length of reads and whether they were paired- or single-end. |
| Antibodies | Describe the antibodies used for the ChIP-seq experiments; as applicable, provide supplier name, catalog number, clone name, and lot number. |

| Peak calling parameters | *Specify the command line program and parameters used for read mapping and peak calling, including the ChIP, control and index files used.* |
|---|---|
| Data quality | *Describe the methods used to ensure data quality in full detail, including how many peaks are at FDR 5% and above 5-fold enrichment.* |
| Software | *Describe the software used to collect and analyze the ChIP-seq data. For custom code that has been deposited into a community repository, provide accession details.* |

# Flow Cytometry

## Plots

Confirm that:

☐ The axis labels state the marker and fluorochrome used (e.g. CD4-FITC).

☐ The axis scales are clearly visible. Include numbers along axes only for bottom left plot of group (a 'group' is an analysis of identical markers).

☐ All plots are contour plots with outliers or pseudocolor plots.

☐ A numerical value for number of cells or percentage (with statistics) is provided.

## Methodology

| Sample preparation | *Describe the sample preparation, detailing the biological source of the cells and any tissue processing steps used.* |
|---|---|
| Instrument | *Identify the instrument used for data collection, specifying make and model number.* |
| Software | *Describe the software used to collect and analyze the flow cytometry data. For custom code that has been deposited into a community repository, provide accession details.* |
| Cell population abundance | *Describe the abundance of the relevant cell populations within post-sort fractions, providing details on the purity of the samples and how it was determined.* |
| Gating strategy | *Describe the gating strategy used for all relevant experiments, specifying the preliminary FSC/SSC gates of the starting cell population, indicating where boundaries between "positive" and "negative" staining cell populations are defined.* |

☐ Tick this box to confirm that a figure exemplifying the gating strategy is provided in the Supplementary Information.

# Magnetic resonance imaging

## Experimental design

| Design type | *Indicate task or resting state; event-related or block design.* |
|---|---|
| Design specifications | *Specify the number of blocks, trials or experimental units per session and/or subject, and specify the length of each trial or block (if trials are blocked) and interval between trials.* |
| Behavioral performance measures | *State number and/or type of variables recorded (e.g. correct button press, response time) and what statistics were used to establish that the subjects were performing the task as expected (e.g. mean, range, and/or standard deviation across subjects).* |

## Acquisition

| Imaging type(s) | *Specify: functional, structural, diffusion, perfusion.* |
|---|---|
| Field strength | *Specify in Tesla* |
| Sequence & imaging parameters | *Specify the pulse sequence type (gradient echo, spin echo, etc.), imaging type (EPI, spiral, etc.), field of view, matrix size, slice thickness, orientation and TE/TR/flip angle.* |
| Area of acquisition | *State whether a whole brain scan was used OR define the area of acquisition, describing how the region was determined.* |
| Diffusion MRI | ☐ Used ☐ Not used |

## Preprocessing

| Preprocessing software | *Provide detail on software version and revision number and on specific parameters (model/functions, brain extraction, segmentation, smoothing kernel size, etc.).* |
|---|---|

| Normalization | *If data were normalized/standardized, describe the approach(es): specify linear or non-linear and define image types used for transformation OR indicate that data were not normalized and explain rationale for lack of normalization.* |
|---|---|
| Normalization template | *Describe the template used for normalization/transformation, specifying subject space or group standardized space (e.g. original Talairach, MNI305, ICBM152) OR indicate that the data were not normalized.* |
| Noise and artifact removal | *Describe your procedure(s) for artifact and structured noise removal, specifying motion parameters, tissue signals and physiological signals (heart rate, respiration).* |
| Volume censoring | *Define your software and/or method and criteria for volume censoring, and state the extent of such censoring.* |

## Statistical modeling & inference

| Model type and settings | *Specify type (mass univariate, multivariate, RSA, predictive, etc.) and describe essential details of the model at the first and second levels (e.g. fixed, random or mixed effects; drift or auto-correlation).* |
|---|---|
| Effect(s) tested | *Define precise effect in terms of the task or stimulus conditions instead of psychological concepts and indicate whether ANOVA or factorial designs were used.* |

Specify type of analysis: ☐ Whole brain  ☐ ROI-based  ☐ Both

| Statistic type for inference<br>(See Eklund et al. 2016) | *Specify voxel-wise or cluster-wise and report all relevant parameters for cluster-wise methods.* |
|---|---|
| Correction | *Describe the type of correction and how it is obtained for multiple comparisons (e.g. FWE, FDR, permutation or Monte Carlo).* |

## Models & analysis

| n/a | Involved in the study |
|---|---|
| ☐ ☐ | Functional and/or effective connectivity |
| ☐ ☐ | Graph analysis |
| ☐ ☐ | Multivariate modeling or predictive analysis |

| Functional and/or effective connectivity | *Report the measures of dependence used and the model details (e.g. Pearson correlation, partial correlation, mutual information).* |
|---|---|
| Graph analysis | *Report the dependent variable and connectivity measure, specifying weighted graph or binarized graph, subject- or group-level, and the global and/or node summaries used (e.g. clustering coefficient, efficiency, etc.).* |
| Multivariate modeling and predictive analysis | *Specify independent variables, features extraction and dimension reduction, model, training and evaluation metrics.* |

