## [Peer Review File · Nature Human Behaviour]

Information transfer within and between autistic and non-autistic people: A registered report

Corresponding Author: Dr Catherine Crompton

Version 0:

Decision Letter:

4th April 2022

Dear Catherine,

Thank you once again for your manuscript, entitled "Information transfer within and between autistic and non-autistic people: a replication study and extensions," and for your patience during the peer review process.

I want to apologize again for the excessive delay in communicating a decision on your submission. As previously discussed, we lost one referee after a considerable wait, and finding an expert able to take their place took us longer than I would have hoped.

Your manuscript has now been evaluated by 2 reviewers, whose comments are included at the end of this letter. Although the reviewers find your protocol to be of interest, they also raise some important concerns. We are very interested in the possibility of proceeding further with your submission in Nature Human Behaviour, but would like to consider your response to these concerns in the form of a revised manuscript before we make a decision on in principle acceptance and Stage 2 submission.

To guide the scope of the revisions, the editors discuss the referee reports in detail within the team, including with the chief editor, with a view to (1) identifying key priorities that should be addressed in revision and (2) overruling referee requests that are deemed beyond the scope of the current study. We hope that you will find the prioritised set of referee points to be useful when revising your study. Please do not hesitate to get in touch if you would like to discuss these issues further.

We ask you to address all of the referees suggestions for improvement and extension of the present paradigm, to ensure the resulting data will be as reliable and interpretable as possible, and the project reaches its full potential to inform a broad audience of researchers.

We also ask you to revise your sampling plan. At 0.9 power, the study is presently underpowered by the standards we maintain across all Registered Reports. For frequentist analysis plans, the a priori power must be 0.95 or higher for all proposed hypothesis tests. A resubmission using a Neyman-Pearson inference based power analysis to determine its sample would need to commit to 0.95 power to be taken forward.

In sum, we invite you to revise your Stage 1 Registered Report taking into account reviewer and editor comments. Please highlight all changes in the manuscript text file.

* Include a "Response to reviewers" document detailing, point-by-point, how you addressed each referee comment. If no action was taken to address a point, you must provide a compelling argument. This response will be sent back to the reviewers along with the revised manuscript.

* Ensure that you use our template for Stage 1 Registered Reports to prepare your revised manuscript: https://www.nature.com/documents/NHB_Template_RR_Stage1.docx. Failure to ensure that your revised Stage 1 submission meets our requirements as specified in the template will result in your submission being returned to you, which will delay its consideration.

* In your cover letter, please include the following information:

--An anticipated timeline for completing the study if your Stage 1 submission is accepted in principle.

--A statement confirming that you agree to share your raw data, any digital study materials, computer code (if relevant), and laboratory log for all eventually published results.

--A statement confirming that, following Stage 1 in principle acceptance, you agree to register your approved protocol on the Open Science Framework (<https://osf.io/>) or other recognised repository, either publicly or under private embargo, until submission of the Stage 2 manuscript.

--A statement confirming that if you later withdraw your paper, you agree to the Journal publishing a short summary of the pre-registered study under a section Withdrawn Registrations.

Link Redacted

We hope to receive your revised manuscript within four to eight weeks. If you cannot send it within this time, please let us know. We will be happy to consider your revision so long as nothing similar has been accepted for publication at Nature Human Behaviour or published elsewhere.

Nature Human Behaviour is committed to improving transparency in authorship. As part of our efforts in this direction, we are now requesting that all authors identified as 'corresponding author' on published papers create and link their Open Researcher and Contributor Identifier (ORCID) with their account on the Manuscript Tracking System (MTS), prior to acceptance. ORCID helps the scientific community achieve unambiguous attribution of all scholarly contributions. You can create and link your ORCID from the home page of the MTS by clicking on 'Modify my Springer Nature account'. For more information please visit <http://www.springernature.com/orcid>.

Sincerely,

[Redacted]

[Redacted]

[Redacted]
Nature Human Behaviour

Reviewer expertise:

Reviewer #1: ASD, language

Reviewer #2: ASD, language

Reviewers' Comments:

Reviewer #1:

Remarks to the Author:

Significance

The original study being replicated was very significant and impactful. It used an original method (transmission chains) to demonstrate that transmission is only substantially impaired in mixed chains (autistic and non-autistic people) compared to homogenous chains (either autistic or non-autistic people). This is significant because it shifts focus from a deficit model of autism toward a relational/communication model.

A key question is whether the proposed research is comparably significant. The proposed research is a replication with a larger sample across three locations. This is welcome, but, given that there is already converging evidence on the original finding, it is arguably important but of limited significance. The proposed research extends the original study in two ways (not "several" – p.4, line 129): (1) it adds a condition for fiction/non-fiction material and (2) disclosing/not disclosing autism. These additional manipulations are welcome, somewhat predictable, and moderately significant.

Nevertheless, the research is clever, socially significant, and likely to be of interest to a broad interdisciplinary audience.

Methodology

The research design, proposed methods, data collection procedures, and plan for the analysis are rigorous, clearly presented, and enable replication. They leave few degrees of freedom, thus indicative of a robust replication. One limitation with the methodology is that rapport is assessed only with self-report, and may thus pick up on different response biases between autistic and non-autistic participants. Ideally, this would be backed up by an observational measure of rapport, such as an independent judge, a sentiment algorithm on the interaction transcripts, or emotion recognition using video.

I suspect that the research will be replicated in part because any diversity introduced into a transmission chain will tend to degrade the original message, as the message requires more extensive translation into different frames of reference. One would expect the same results for chains that mixed nationality, age, or social class etc. In a sense, this transmission paradigm (with the outcome being accuracy) could be interpreted as a measure of the extent to which the members of the chain share assumptions, communication heuristics, and frames of reference. The research states that participants will be matched for age, gender, and IQ. I wonder if there should also be some matching for social class, education level, and linguistic ability.

Reviewer #2:

Remarks to the Author:

In this Registered Report, the authors report on the use of a diffusion chain method to examine information transfer in groups of autistic people, groups of non-autistic people, and mixed groups of autistic and non-autistic people.

The research questions are important and relevant to a broad, multidisciplinary audience, and the proposed hypotheses are logic and plausible. The methodology and analysis pipeline (including statistical power analysis) are appropriate and provided in sufficient detail for replica.

Suggestions for improvement:

1) Transmission of information. As in Crompton et al., 2020, "a participant's final score will correspond to the number of details they recall when recounting the passage to the next person in the chain, out of a maximum of 30". Recall (number of details transmitted out of a maximum of 30) measures the quantity of information transmitted to the next person. In addition, the authors may want to measure the fidelity (accuracy) with which the message content is transmitted. This could be done, for example, by calculating the number of distortions and the number of new details that are generated. An example of this approach can be found in: <https://doi.org/10.1073/pnas.1421883112>

2) Mismatch model of ASD. The authors derive their predictions from a mismatch-model of ASD's difficulties in social interaction. This model has received empirical support from studies showing that kinematic similarity is important for action prediction. ASD observers are better able to read/predict ASD actions than TD actions (<https://doi.org/10.1073/pnas.2114648119>) and conversely, TD observers are better able to read/predict TD actions than ASD actions (<https://doi.org/10.1073/pnas.2114648119>; <https://doi.org/10.1073/pnas.2011311117>).

The authors may want to refer to these studies as supporting a kinematic dissimilarity hypothesis of ASD difficulties in social interaction (<https://doi.org/10.1098/rstb.2015.0372>).

In this view, the difficulties in social interaction would not arise because of social judgements, and misunderstandings of non-autistic social partners (a "deficit" on the side of non-autistic social partners) but because of objective differences in the way TD and ASD communicate.

Version 1:

Decision Letter:

11th May 2022

Dear Catherine,

Thank you for submitting your revised manuscript, "Information transfer within and between autistic and non-autistic people: a replication study and extensions". After careful consideration and discussion with my colleagues, I am sorry to have to tell you that we do not feel that the referees' and editors' comments have been sufficiently addressed to justify sending this revision back for peer review.

This unusual course of action is taken occasionally to avoid unproductive rounds of review that result in reviewer fatigue and damage the chances of the manuscript obtaining a fair and objective evaluation. Such situations are not in an author's best interest so we try to avoid them when it seems prudent to do so.

In order to consider this manuscript further we would request that you please do your best to fully address all of the comments of the reviewers and the editorial requests. In particular, please:

- Provide a sampling plan based on a power-analysis that complies with our guidelines. It is unclear how the effect size was chosen. There should be a justification for why the targeted effect size is the smallest effect of theoretical or practical significance, for which $\eta^2 = 0.24$ appears a very large value.
- Respond to the referees' concerns by implementing the necessary changes in the protocol, rather than rebutting their points of critique without undertaking any changes to the design.
- Do not remove hypotheses from the manuscript. Instead, the work must be sufficiently powered (95%) to test the relevant hypotheses.

Should you be able to adequately respond to these and the reviewers' other concerns, we would be happy to look at a revised manuscript.

We shall hope to receive your revised version as soon as possible. If you anticipate a delay of more than four weeks, however, please let us know. We will be happy to consider your revision so long as nothing similar has been accepted for publication at

Nature Human Behaviour or published elsewhere. Should your manuscript be substantially delayed without notifying us in advance and your article is eventually published, the received date may be that of the revised, not the original, version.

If you are not interested in submitting a suitably revised manuscript in the future please let me know immediately so we can close your file. If you have any questions, please contact me.

Please use the link below to submit a suitably revised manuscript and updated response to referees when they are ready.

Link Redacted

Sincerely,

[Redacted]

[Redacted]

Nature Human Behaviour

Version 2:

Decision Letter:

23rd August 2022

Dear Dr Fletcher-Watson,

Thank you once again for your Stage 1 Registered Report manuscript, entitled "Information transfer within and between autistic and non-autistic people: a replication study and extensions" and for your patience during the peer review process.

Your manuscript has now been evaluated again by 3 reviewers, including new statistics referee #4, whose comments are included at the end of this letter. The reviewers all feel that the revised manuscript is much improved, and are happy to recommend acceptance of the Stage 1 protocol.

However, Reviewer #4 raises some comments that should be addressed. In addition, before we can finally accept this manuscript, we would like you to make some further changes to the text to comply with our formatting requirements.

I have attached a Checklist document which gives guidance on formatting of Stage 1 Registered Reports. I have added a number of comments in the Word file, which contain my instructions on the most important changes that must be made. Please let me know if you have any difficulty in accessing these comments.

In sum, we invite you to revise your manuscript taking into account all reviewer and editor comments. We are committed to providing a fair and constructive peer-review process. Do not hesitate to contact us if there are specific requests from the reviewers that you believe are technically impossible or unlikely to yield a meaningful outcome.

We hope to receive your revised manuscript within 4 weeks. I would be grateful if you could contact us as soon as possible if you foresee difficulties with meeting this target resubmission date.

- Include a "Response to the editors and reviewers" document detailing, point-by-point, how you addressed each editor and referee comment. If no action was taken to address a point, you must provide a compelling argument. When formatting this document, please respond to each reviewer comment individually, including the full text of the reviewer comment verbatim followed by your response to the individual point. This response will be used by the editors to evaluate your revision and sent back to the reviewers along with the revised manuscript.
- Highlight all changes made to your manuscript or provide us with a version that tracks changes.

Link Redacted

We look forward to seeing the revised manuscript and thank you for the opportunity to review your work. Please do not hesitate to contact me if you have any questions or would like to discuss these revisions further.

Sincerely,

[Redacted]

[Redacted]

REVIEWER COMMENTS:

Reviewer #2:

Remarks to the Author:

This revised proposal is improved and satisfactorily addresses the issues raised. The extra details improve the rigor, the additional controls for education etc will make the analysis more robust, and the rationale for using self-report is reasonable. On this basis, I'm happy to support this as a registered report.

Reviewer #3:

Remarks to the Author:

The authors have made a serious effort to address all my comments. I find the manuscript to be greatly improved and look forward to the results of this study.

Reviewer #4:

Remarks to the Author:

I was tasked with evaluating the report's sample size and power analysis. The authors need to update a few things, but with such a large sample size, they will likely be sufficiently powered for finding medium effects.

1. The sample size seems appropriate, but clarification is necessary about the model chosen for the simulation-based power analysis. Here, the authors are using an intercept-only model ("1 | chain"), but no theoretical justification for the random effects structure is given. Choosing intercept-only models out-of-the-box, without a strong theoretical reason, can inflate the probability of alpha errors and can reduce generalisability:

Barr DJ, Levy R, Scheepers C, Tily HJ. Random effects structure for confirmatory hypothesis testing: Keep it maximal. *J Mem Lang*. 2013 Apr;68(3):10.1016/j.jml.2012.11.001.

Either choosing the "maximal" random effect structure (probably "1 + position | chain" if I understood the within/between manipulations correctly) is necessary, or following very careful and conservative random effects-simplification is required:

Bates D., Kliegl R., Vasishth S., & Baayen H. Parsimonious mixed models. *arXiv*. 2015; (arXiv:1506.04967v2)
<https://arxiv.org/abs/1506.04967>

Either way, this needs to be justified and/or the power analysis needs to be re-run with the justified random-effects structure. Most likely the selected sample size will nevertheless be sufficiently large.

2. The authors state that the simulations "are based on reduced coefficients", but it is not transparent how much they were reduced. Providing a % would be important.

3. Finally, the OSF link (<https://osf.io/us9c7/>) did not lead to R code. The authors state "Further details of the power analysis and simulations in R can be found in the R-file", so this needs to be updated.

Version 3:

Decision Letter:

5th October 2022

Dear Professor Fletcher-Watson,

Thank you once again for submitting your revised Stage 1 Registered Report, entitled "Information transfer within and between autistic and non-autistic people." Everything is in order and I am delighted to say that we can offer acceptance in principle. You may progress to Stage 2 and complete the study as approved.

As you know, a condition of in-principle-acceptance is that the authors agree to deposit their Stage 1 accepted protocol in a repository, either publicly or under embargo until Stage 2 acceptance and publication. We are very keen to showcase our in-

principle accepted protocols, so that our readers, reviewers, and potential authors can gain insight into the requirements of the format as well as an idea of the types of projects that are suitable for publication in Nature Human Behaviour.

We have set up a space on figshare (https://springernature.figshare.com/registered-reports_NHB) to host all of our in-principle accepted protocols, which can either be made public or kept under embargo until Stage 2 acceptance (depending on author preference). This gives you the opportunity to have your work publicly associated with Nature Human Behaviour, and of course we will be very pleased to showcase your report if you agree to share it publicly.

Depositing the work on our figshare space does not preclude deposition of your Stage 1 protocol on other depositories – your protocol can also be posted on OSF, Dataverse, Dryad or any other public repository of your choice. You also do not need to do anything – if you agree with posting your protocol on our figshare space, we will upload your protocol on your behalf and either set it public or place it under embargo, depending on your choice.

Your protocol will be licensed under a CC BY license (Creative Commons Attribution 4.0 International License). The CC BY license allows for maximum dissemination and re-use of open access materials and is preferred by many research funding bodies. Under this license users are free to share (copy, distribute and transmit) and remix (adapt) the contribution including for commercial purposes, providing they attribute the contribution in the manner specified by the author or licensor (read full legal code: <http://creativecommons.org/licenses/by/4.0/legalcode>) Please note that any use of <https://springernature.figshare.com> will be subject to the Figshare terms of use. Figshare has the right to enforce these terms and conditions where applicable. Use of third party services and sites will be subject to the relevant terms of use and will apply if we act on your behalf in this regard.

Do let me know if you would like to take up this option or if you have any questions regarding the protocol deposition requirement.

Following completion of your study, we invite you to resubmit your paper for peer review as a Stage 2 Registered Report. Please note that your manuscript can still be rejected for publication at Stage 2 if the Editors consider any of the following to hold:

- The results were unable to test the authors' proposed hypotheses by failing to meet the approved outcome-neutral criteria
- The authors altered the Introduction, rationale, or hypotheses, as approved in the Stage 1 submission
- The authors failed to adhere closely to the registered experimental procedures
- Any post hoc (unregistered) analyses were either unjustified, insufficiently caveated, or overly dominant in shaping the authors' conclusions
- The authors' conclusions were not justified given the data obtained

We encourage you to read the complete guidelines for authors concerning Stage 2 submissions at <https://www.nature.com/nathumbehav/registeredreports>. Please especially note the requirements for protocol deposition, data sharing, and that withdrawing your manuscript will result in publication of a Retracted Registration.

In recognition of the time and expertise our reviewers provide to Nature Human Behaviour's editorial process, we would like to formally acknowledge their contribution to the external peer review of your manuscript entitled "Information transfer within and between autistic and non-autistic people". For those reviewers who give their assent, we will be publishing their names alongside the published article.

When you are ready, please use the following link to access your home page and submit your Stage 2 Registered Report:

Link Redacted

*This url links to your confidential homepage and associated information about manuscripts you may have submitted or be reviewing for us. If you wish to forward this e-mail to co-authors, please delete this link to your homepage first.

We expect your Stage 2 Registered Report to be submitted by the date specified in your latest cover letter. If unforeseen circumstances prevent submission by that date, please contact us as soon as possible to discuss any changes to the submission time-frame.

Thank you again for offering us this work and we look forward to receiving your Stage 2 Registered Report.

Yours sincerely,

Nature Human Behaviour

Version 4:

Decision Letter:

21st January 2025

Dear Dr Crompton,

Apologies for the confusion. I realized that we had not provided you with an open link for the re-submission of your revised manuscript. You can now find it here below.

As mentioned in our prior email exchange, we are pleased to inform you that we will be able accept your Stage 2 manuscript, pending revisions to address reviewer comments and editorial requests.

Please attend to **every item** in the checklist and upload a copy of the completed checklist with your submission. I also mention here a few points that are frequently missed and can cause delays:

- 1) Please insert the following Protocol Registration information in your manuscript:
- 2) Ensure that all corresponding authors have linked their ORCID to their account on our online manuscript handling system. This is very frequently missed and invariably causes delays in formal acceptance.
- 3) Ensure that you provide all of the materials requested in the attached checklist and below with your final submission.

Nature Human Behaviour offers a transparent peer review option for new original research manuscripts submitted from 1st December 2019. We encourage increased transparency in peer review by publishing the reviewer comments, author rebuttal letters and editorial decision letters if the authors agree. Such peer review material is made available as a supplementary peer review file. **Please state in the cover letter 'I wish to participate in transparent peer review' if you want to opt in, or 'I do not wish to participate in transparent peer review' if you don't.** Failure to state your preference will result in delays in accepting your manuscript for publication.

To submit your revised manuscript, you will need to provide the following:

- Cover letter
- Point-by-point response to the reviewers (if applicable)
- Manuscript text (not including the figures) in .docx or .tex format
- Individual figure files (one figure per file)
- Extended Data & Supplementary Information, as instructed
- Reporting summary
- Editorial policy checklist
- Third-party rights table (if applicable)
- Suggestions for cover illustrations (if desired)

Consortia authorship:

For papers containing one or more consortia, all members of the consortium who contributed to the paper must be listed in the paper (i.e., print/online PDF). If necessary, individual authors can be listed in both the main author list and as a member of a consortium listed at the end of the paper. When submitting your revised manuscript via the online submission system, the consortium name should be entered as an author, together with the contact details of a nominated consortium representative. See <https://www.nature.com/authors/policies/authorship.html> for our authorship policy and <https://www.nature.com/documents/nr-consortia-formatting.pdf> for further consortia formatting guidelines, which should be adhered to prior to acceptance.

Forms:

Nature Human Behaviour has now transitioned to a unified Rights Collection system which will allow our Author Services team to quickly and easily collect the rights and permissions required to publish your work. Once your paper is accepted, you will receive an email in approximately 10 business days providing you with a link to complete the grant of rights. If you choose to publish Open Access, our Author Services team will also be in touch at that time regarding any additional information that may be required to arrange payment for your article.

For information regarding our different publishing models please see our <https://www.springernature.com/gp/open-research/transformational-journals> page. If you have any questions about costs, Open Access requirements, or our legal forms, please contact ASJournals@springernature.com.

Link Redacted

With best regards,

Nature Human Behaviour

Reviewer #3 (Remarks to the Author):

Ability of the Data to Test Proposed Hypotheses. The data collected appears suitable for testing the stated hypotheses. The study effectively measured information transfer and rapport, aligning with the hypotheses outlined. Quality checks, such as the reliability of coding, were reported to be robust, supporting the validity of the findings.

Consistency of Introduction, Rationale, and Stated Hypotheses. The Introduction and stated hypotheses in the submission remain consistent with those in the pre-registration. The study builds on previous findings, aiming to replicate and extend the understanding of information transfer and rapport in interactions involving matched and mixed neurotypes. The core hypotheses regarding information decay and rapport differences are unchanged.

Adherence to Registered Experimental Procedures. The study adhered closely to the experimental procedures outlined in the pre-registration. The diffusion chain method was implemented as planned, and participant criteria were followed rigorously. A minor deviation occurred where one analysis could not be completed due to reliability issues, but this was transparently reported and justified.

Unregistered Post Hoc Analyses. The submission includes post hoc analyses that are methodologically sound and justified by the data. These additional analyses, such as controlling for demographic variables and exploring non-linear decay patterns, enhance the robustness of the findings without deviating from the study's aims.

Justification of Conclusions. The authors' conclusions are well-supported by the data. The study confirmed some of the expected outcomes, such as differences in rapport scores, while other predictions, like selective information breakdown in mixed chains, were not replicated. The conclusions acknowledge the potential influence of sample diversity and methodological choices.

Version 5:

Decision Letter:

Dear Dr Crompton,

We are pleased to inform you that your Registered Report "Information transfer within and between autistic and non-autistic people: A registered report", has now been accepted for publication in Nature Human Behaviour.

To assist our authors in disseminating their research to the broader community, our SharedIt initiative provides you with a unique shareable link that will allow anyone (with or without a subscription) to read the published article. Recipients of the link with a

subscription will also be able to download and print the PDF.

With best regards,

Nature Human Behaviour

P.S. Click on the following link if you would like to recommend Nature Human Behaviour to your librarian
<http://www.nature.com/subscriptions/recommend.html#forms>

** Visit the Springer Nature Editorial and Publishing website at http://editorial-jobs.springernature.com?utm_source=ejp_NHumB_email&utm_medium=ejp_NHumB_email&utm_campaign=ejp_NHumB for more information about our career opportunities. If you have any questions please click [here](mailto:editorial.publishing.jobs@springernature.com).

We are really grateful for the thoughtful and constructive feedback of the reviewers. We have addressed each of the points the reviewers raised and have edited the manuscript accordingly. We hope that our revised manuscript answers all the comments. We have made a serious effort to address these comments, and as a result, we believe the manuscript to be greatly improved.

Editor comments

We also ask you to revise your sampling plan. At 0.9 power, the study is presently underpowered by the standards we maintain across all Registered Reports. For frequentist analysis plans, the a priori power must be 0.95 or higher for all proposed hypothesis tests. A resubmission using a Neyman-Pearson inference based power analysis to determine its sample would need to commit to 0.95 power to be taken forward.

Our analyses suggest that planned frequentist significance tests of the main hypotheses can be conducted with power of 95% or higher (R-packages *pwr*, *simr*). However, it is correct that some interactions where we anticipate small effect sizes and plan to use frequentist testing are likely to be underpowered, against the described journal standard. We have therefore removed Hypotheses 1c and 2c (two-way interaction of chain type and position) from the predicted effects. They are not essential to the main goal of our study, and have now been included as post-hoc analyses instead. Additionally, concerns about power do not apply to many of our analyses where we will apply Bayesian inferences, which are better suited to investigate interactions and non-linear effects of dependent variable information loss and rapport score over time.

We have made small edits to the text to reflect this, which are shown in red text in the manuscript.

Reviewer #1:

Remarks to the Author:

Significance

1. The original study being replicated was very significant and impactful. It used an original method (transmission chains) to demonstrate that transmission is only substantially impaired in mixed chains (autistic and non-autistic people) compared to homogenous chains (either autistic or non-autistic people). This is significant because it shifts focus from a deficit model of autism toward a relational/communication model.

A key question is whether the proposed research is comparably significant. The proposed research is a replication with a larger sample across three locations. This is welcome, but, given that there is already converging evidence on the original finding, it is arguably important but of limited significance. The proposed research extends the original study in two ways (not “several” – p.4, line 129): (1) it adds a condition for fiction/non-fiction material and (2) disclosing/not disclosing autism. These additional manipulations are welcome, somewhat predictable, and moderately significant.

Nevertheless, the research is clever, socially significant, and likely to be of interest to a broad interdisciplinary audience.

Thank you for your helpful and constructive comments, we really appreciate the time you have taken to provide them and your generous comments about the significance and broad interest of the research. We agree! And we also agree that we could be clearer about the proposed extensions to the study, and as per your suggestion have amended “several” to “two”.

2. The research design, proposed methods, data collection procedures, and plan for the analysis are rigorous, clearly presented, and enable replication. They leave few degrees for freedom, thus indicative of a robust replication.

One limitation with the methodology is that rapport is assessed only with self-report, and may thus pick up on different response biases between autistic and non-autistic participants. Ideally, this would be backed up by an observational measure of rapport, such as an independent judge, a sentiment algorithm on the interaction transcripts, or emotion recognition using video.

Thank you for your support of our research design, methods, procedures and analysis plan.

The self-rated rapport tool we use is designed to allow participants to report on their interactional rapport with each person adjacent to them in the diffusion chain. Studies of rapport in dyadic interactions typically use self-rated questionnaires (e.g. Frisby and Martin, 2010), and while it is possible that these may be subject to response biases, it is the most direct way to assess each participants' experience of the interaction.

We have previously examined both self-rated and observer-rated rapport in autistic, non-autistic, and mixed dyads (Crompton et al., 2020), and found that observations of rapport typically align with self-rated rapport: mixed pairs being rated as having lower rapport than single-neurotype dyads.

It is possible that both autistic and non-autistic people have a response bias when self-rating rapport: in the original study, we found that autistic dyads experience lower rapport than non-autistic dyads. It may be the case that autistic people under-estimate their rapport (perhaps due to negative self-perception of their social skills, lower perceived social competence, or a history of difficult social interactions with others), or that non-autistic people over-estimate their rapport. However, we are specifically interested in whether rapport varies depending on social context (if people are in a single-neurotype or mixed dyad) rather than as a main effect of diagnosis (autistic or non-autistic). In this context, we think that the influence of any response bias that varies between diagnostic groups is well-managed by the study design.

Additionally, independent observational ratings are quite weak for capturing neurodiverse interactional experiences. Specifically, they can be biased by a neuro-normative lens, and thus we opted not to include them in this work. For example, when independent judges rate videos of autistic people, they rate them as being more awkward and less approachable (Sasson et al., 2017), both of which are key factors in building rapport. These biases are robust, are developed very rapidly, and do not change with increased exposure (Sasson et al., 2017). Importantly, autistic independent judges have a similar tendency to non-autistic judges to evaluate autistic adults less favourably than non-autistic adults in videos (DeBrabander et al., 2019), and so this bias cannot be overcome by simply recruiting both autistic and non-autistic independent judges. Similarly, emotion recognition is subject to strong neuro-normative biases – autistic people have different facial expressions to non-autistic people, and non-autistic people (and thus, emotion recognition software based on non-autistic norms) are poor at identifying these emotions (Sheppard et al., 2016), which is related to their perceiving them unfavourably. Normative biases are also built into sentient algorithms (Fletcher-Watson et al., 2018), and thus automated tools based in machine learning are similarly problematic to use in this context. For these reasons, we believe that the most valid measure is to use self-report measures, as in the original study.

That said, we of course recognise the high value and broad interest of the data we will be collecting in this study, and are committed to sharing these data, within participant consent, for future examination. Exploratory work looking at alternative ways to capture interactional dynamics will doubtless form a key part of these exploratory extensions. Such analyses of the raw data collected in this study (e.g. from human or automated coding) fall outwith the scope of this report.

References

- Alkhaldi, R. S., Sheppard, E., & Mitchell, P. (2019). Is there a link between autistic people being perceived unfavorably and having a mind that is difficult to read?. *Journal of Autism and Developmental Disorders*, 49(10), 3973-3982.
- Crompton, C. J., Sharp, M., Axbey, H., Fletcher-Watson, S., Flynn, E. G., & Ropar, D. (2020). Neurotype-matching, but not being autistic, influences self and observer ratings of interpersonal rapport. *Frontiers in psychology*, 2961.
- DeBrabander, K. M., Morrison, K. E., Jones, D. R., Faso, D. J., Chmielewski, M., & Sasson, N. J. (2019). Do first impressions of autistic adults differ between autistic and nonautistic observers?. *Autism in Adulthood*, 1(4), 250-257.

- Fletcher-Watson, S., De Jaegher, H., Van Dijk, J., Frauenberger, C., Magnée, M., & Ye, J. (2018). Diversity computing. *Interactions*, 25(5), 28-33.
- Frisby, B. N., and Martin, M. M. (2010). Instructor-student and student-student rapport in the classroom. *Commun. Educ.* 59, 146–164. doi: 10.1080/03634520903564362
- Sasson, N. J., Faso, D. J., Nugent, J., Lovell, S., Kennedy, D. P., & Grossman, R. B. (2017). Neurotypical peers are less willing to interact with those with autism based on thin slice judgments. *Scientific reports*, 7(1), 1-10.
- Sheppard, E., Pillai, D., Wong, G. T. L., Ropar, D., & Mitchell, P. (2016). How easy is it to read the minds of people with autism spectrum disorder?. *Journal of autism and developmental disorders*, 46(4), 1247-1254.

3. I suspect that the research will be replicated in part because any diversity introduced into a transmission chain will tend to degrade the original message, as the message requires more extensive translation into different frames of reference. One would expect the same results for chains that mixed nationality, age, or social class etc. In a sense, this transmission paradigm (with the outcome being accuracy) could be interpreted as a measure of the extent to which the members of the chain share assumptions, communication heuristics, and frames of reference. The research states that participants will be matched for age, gender, and IQ. I wonder if there should also be some matching for social class, education level, and linguistic ability.

Thank you for raising this interesting point. We agree that demonstrating a specific effect of neurotype will be more powerful if we can capture and match for as many other factors as possible.

To incorporate your recommendations relating to social class we will ask participants to provide details of parental education level (both as years of education, and highest educational level achieved), a proxy for socio-economic status (Aarø et al., 2019) which can be used to compare across participants in all groups at all sites.

Additionally, we will ask participants to provide details of their education level (both as years of education, and highest educational level achieved).

To measure IQ, we are using the Wechsler Abbreviated Scale of Intelligence II (WASI-II), which includes a verbal IQ measure. Thus, by ensuring groups are comparable on their WASI-II scores, we are matching for their linguistic ability.

We will use these variables to match groups as far as possible. If it is not possible to match groups, we will examine the possible impact of this by entering the variable as a co-variate in a post-hoc analysis

We have now included the following information in-text relating to this change:

We will attempt to ensure that groups are comparable on gender, age, educational level, IQ, and socio-economic status (as indicated by parent education level (Aarø et al., 2019)). If it is not possible to match groups, we will examine the possible impact of this by entering the variable as a co-variate in a post-hoc analysis.

New references added

Aarø, L. E., Flisher, A. J., Kaaya, S., Onya, H., Namisi, F. S., & Wubs, A. (2009). Parental education as an indicator of socioeconomic status: improving quality of data by requiring consistency across measurement occasions. *Scandinavian journal of public health*, 37(2_suppl), 16-27.

Reviewer #2:

Remarks to the Author:

In this Registered Report, the authors report on the use of a diffusion chain method to examine information transfer in groups of autistic people, groups of non-autistic people, and mixed groups of autistic and non-autistic people. The research questions are important and relevant to a broad, multidisciplinary audience, and

the proposed hypotheses are logic and plausible. The methodology and analysis pipeline (including statistical power analysis) are appropriate and provided in sufficient detail for replica.

Suggestions for improvement:

1) Transmission of information. As in Crompton et al., 2020, “a participant’s final score will correspond to the number of details they recall when recounting the passage to the next person in the chain, out of a maximum of 30”. Recall (number of details transmitted out of a maximum of 30) measures the quantity of information transmitted to the next person. In addition, the authors may want to measure the fidelity (accuracy) with which the message content is transmitted. This could be done, for example, by calculating the number of distortions and the number of new details that are generated. An example of this approach can be found in: <https://doi.org/10.1073/pnas.1421883112>

Thank you for your review and for the suggestions you have made to improve our manuscript.

We agree that examining the number of distortions/new details that occur within chains could be an interesting way to look at information transfer within groups, and appreciate this suggestion. As we have no way to predict what the distribution of this variable may be, we have included this within the Exploratory Analysis section. We have proposed formulating the variable, checking the distribution, and if it is sufficient for analysis, we will repeat a similar analysis to that for Hypothesis 1. We have included the following detail outlining this new proposed analysis:

Exploring Distortions in Diffusion Chain Content: In diffusion chains, the content of an original piece of information generally degrades over repeated social transmissions (Flynn & Whiten, 2008). However, it is also possible that information becomes distorted, with participants adding detail not included in the original information (e.g. “She turned left at the blue windmill” becomes “She turned left at the blue flowery windmill”) or making an error in information transfer (e.g. “She turned left at the blue windmill” becomes “She turned left at the blue castle”), which is then transferred to subsequent participants (Moussaïd, Brighton & Gaissmaier, 2015). Thus, in addition to counting the number of correct details transmitted, we will also examine transcripts of interactions to quantify distortions occurring in information transfer. We will examine and visualise the data and check the assumptions of the analysis method as described in Hypotheses 1. If the assumptions are met, then we will perform a linear regression analysis, using a simple linear model, and if not, we will apply Bayesian analysis methods. This will include the dependent variable of individual participant distortion score (i.e. how many distortions each participant introduced), and predictor variables of chain type (autistic, non-autistic, mixed) and chain position (1 to 6). Collectively, analyses will therefore determine both if information degrades differently across chain-types and whether new incorrect information is generated to a greater degree in some chain types (and by some participants) compared to others.

New references

Moussaïd, M., Brighton, H., & Gaissmaier, W. (2015). The amplification of risk in experimental diffusion chains. *Proceedings of the National Academy of Sciences*, 112(18), 5631-5636.

2) Mismatch model of ASD. The authors derive their predictions from a mismatch-model of ASD’s difficulties in social interaction. This model has received empirical support from studies showing that kinematic similarity is important for action prediction. ASD observers are better able to read/predict ASD actions than TD actions (<https://doi.org/10.1073/pnas.2114648119>) and conversely, TD observers are better able to read/predict TD actions than ASD actions (<https://doi.org/10.1073/pnas.2114648119>; <https://doi.org/10.1073/pnas.2011311117>).

The authors may want to refer to these studies as supporting a kinematic dissimilarity hypothesis of ASD difficulties in social interaction (<https://doi.org/10.1098/rstb.2015.0372>).

In this view, the difficulties in social interaction would not arise because of social judgements, and misunderstandings of non-autistic social partners (a “deficit” on the side of non-autistic social partners) but because of objective differences in the way TD and ASD communicate.

Thank you so much for this suggestion. We agree that it may be of interest to our readers to include the kinematic dissimilarity theory as a proposed mechanism of interactive differences between autistic and non-autistic people. We have incorporated your suggested literature in the introduction as follows:

This is supported by empirical studies of the kinematic dissimilarity hypothesis of social interaction, which suggests that kinematic similarity is important for action predication and social interaction (Cook, 2016). There are well documented kinematic differences between autistic and non-autistic people (Cook, Blakemore & Press, 2013), and recent studies have found that autistic observers are more able to accurately predict autistic actions than non-autistic actions (Montobbio, et al., 2022), and conversely non-autistic observers are more able to accurately predict non-autistic actions than autistic actions (Cook, 2016). These data suggest that difficulties in social interactions arise because of objective differences in the way that autistic and non-autistic people communicate, rather than an autistic social deficit.

New references

- Cook, J. L., Blakemore, S. J., & Press, C. (2013). Atypical basic movement kinematics in autism spectrum conditions. *Brain*, *136*(9), 2816-2824.
- Cook, J. (2016). From movement kinematics to social cognition: the case of autism. *Philosophical Transactions of the Royal Society B: Biological Sciences*, *371*(1693), 20150372.
- Montobbio, N., Cavallo, A., Albergo, D., Ansuini, C., Battaglia, F., Podda, J., ... & Becchio, C. (2022). Intersecting kinematic encoding and readout of intention in autism. *Proceedings of the National Academy of Sciences*, *119*(5), e2114648119.

We are really grateful for the thoughtful and constructive feedback of the reviewers. We have addressed each of the points the reviewers raised and have edited the manuscript accordingly, with edits highlighted in red. We hope that our revised manuscript answers all the comments. We have made a serious effort to address these comments, and as a result, we believe the manuscript to be greatly improved.

Editor comments

We also ask you to revise your sampling plan. At 0.9 power, the study is presently underpowered by the standards we maintain across all Registered Reports. For frequentist analysis plans, the a priori power must be 0.95 or higher for all proposed hypothesis tests. A resubmission using a Neyman-Pearson inference based power analysis to determine its sample would need to commit to 0.95 power to be taken forward.

Our analyses suggest that planned frequentist significance tests of the main hypotheses can be conducted with power of 95% or higher (R-packages lme4, simr) for a medium effect size in a mixed design (chain type – between, position – within) where number of chains rather than participants determine the sample size. Concerns about power do not apply to our post-hoc analyses where we will apply Bayesian inference, which is better suited to investigate effects of more sophisticated non-linear models.

In the “*Response to additional Editor Feedback*” section below, we highlight in detail the amendments made to our design and power calculations to meet this power threshold.

Reviewer #1:

Remarks to the Author

Significance

1. The original study being replicated was very significant and impactful. It used an original method (transmission chains) to demonstrate that transmission is only substantially impaired in mixed chains (autistic and non-autistic people) compared to homogenous chains (either autistic or non-autistic people). This is significant because it shifts focus from a deficit model of autism toward a relational/communication model.

A key question is whether the proposed research is comparably significant. The proposed research is a replication with a larger sample across three locations. This is welcome, but, given that there is already converging evidence on the original finding, it is arguably important but of limited significance. The proposed research extends the original study in two ways (not “several” – p.4, line 129): (1) it adds a condition for fiction/non-fiction material and (2) disclosing/not disclosing autism. These additional manipulations are welcome, somewhat predictable, and moderately significant.

Nevertheless, the research is clever, socially significant, and likely to be of interest to a broad interdisciplinary audience.

Thank you for your helpful and constructive comments, we really appreciate the time you have taken to provide them and your generous comments about the significance and broad interest of the research. We agree! And we also agree that we could be clearer about the proposed extensions to the study, and as per your suggestion have amended “several” to “two”.

We would also add that, while the reviewer is certainly correct to note that there is converging evidence that autistic people often have more positive social experiences with other autistic people, the specific diffusion chain effect reported in the original study, and more broadly the finding that communication transfer breaks down selectively in interactions between autistic and non-autistic people, has not yet been replicated either directly or conceptually.

2. The research design, proposed methods, data collection procedures, and plan for the analysis are rigorous, clearly presented, and enable replication. They leave few degrees for freedom, thus indicative of a robust replication.

One limitation with the methodology is that rapport is assessed only with self-report, and may thus pick up on different response biases between autistic and non-autistic participants. Ideally, this would be backed up by an observational measure of rapport, such as an independent judge, a sentiment algorithm on the interaction transcripts, or emotion recognition using video.

Thank you for your support of our research design, methods, procedures and analysis plan.

The central aim of this study is to examine whether participants experience rapport with their partner, and if that feeling is mutual. The self-rated rapport tool we propose using allows participants to report on their interactional rapport with each person adjacent to them in the diffusion chain. Studies of rapport in dyadic interactions typically use self-rated questionnaires (e.g. Frisby and Martin, 2010), and while it is possible that these may be subject to response biases, it is the most direct way to assess each participants' direct experience of the interaction. This also replicates the procedure from the original study (Crompton et al., 2020).

Observer ratings of rapport are not central to the research aims of this study: in fact, independent observational ratings are quite weak for capturing neurodiverse interactional experiences. Specifically, they can be biased by a neuro-normative lens: normative external indicators of rapport are less likely to be observed between autistic pairs (Rifai et al., 2022), and this may be undetected or misinterpreted by observers. This means that even if autistic pairs are experiencing high rapport (as we would expect, based on the previous study), external observers are likely to rate them as having low rapport, based on the neurotypical lens embedded in external rapport measures.

For example, when independent judges rate videos of autistic people, they rate them as being more awkward and less approachable (Sasson et al., 2017), both of which are key factors in building rapport. These biases are robust, are developed very rapidly, and do not change with increased exposure (Sasson et al., 2017). Importantly, autistic independent judges have a similar tendency to non-autistic judges to evaluate autistic adults less favourably than non-autistic adults in videos (DeBrabander et al., 2019), and so this bias cannot be overcome by simply recruiting both autistic and non-autistic independent judges. Similarly, emotion recognition is subject to strong neuro-normative biases – autistic people have different facial expressions to non-autistic people, and non-autistic people (and thus, emotion recognition software based on non-autistic norms) are poor at identifying these emotions (Sheppard et al., 2016), which is related to their perceiving them unfavourably. Normative biases are also built into sentient algorithms (Fletcher-Watson et al., 2018), and thus automated tools based in machine learning are similarly problematic to use in this context. For these reasons, we believe that the most valid measure is to use self-report measures, as in the original study.

It is possible that both autistic and non-autistic people have a response bias when self-rating rapport: in the original study, we found that autistic dyads experience lower rapport than non-autistic dyads. It may be the case that autistic people under-estimate their rapport (perhaps due to negative self-perception of their social skills, lower perceived social competence, or a history of difficult social interactions with others), or that non-autistic people over-estimate their rapport. However, we are specifically interested in whether rapport varies depending on dyadic condition (if people are in a single-neurotype or mixed dyad) rather than as a main effect of diagnosis (autistic or non-autistic). In this context, the influence of any response bias that varies between diagnostic groups is well-managed by the study design.

As mentioned above, observer perceptions of rapport are not central to the core aims of this specific study. However, we recognise the high value and broad interest of the data we will be collecting for this study, and that observer ratings of rapport may form a separate research question. We are committed to sharing these data, with participant consent, for future examination. Exploratory work

looking at alternative ways to capture interactional dynamics will doubtless form a key part of these extensions.

In summary, we feel that external rapport ratings of the dyad interactions (from human or automated coding) are subject to normative biases and ask a different research question than the self-rated rapport measures which directly address our research aims. We have added some text to the manuscript explaining our reasons for using self-rated rapport rather than observer rated rapport in this study (pages 8-9, additions indicated in red). However, if the reviewer feels very strongly that external rapport ratings are essential to allow publication of the manuscript, we are willing to consider amending the design to incorporate an external measure. We would appreciate the reviewer's thoughts on which type of external rapport measures (e.g. independent judges (and how many), sentient algorithms, or emotion recognition software) would be most appropriate.

References

- Alkhaldi, R. S., Sheppard, E., & Mitchell, P. (2019). Is there a link between autistic people being perceived unfavorably and having a mind that is difficult to read?. *Journal of Autism and Developmental Disorders*, 49(10), 3973-3982.
- Crompton, C. J., Sharp, M., Axbey, H., Fletcher-Watson, S., Flynn, E. G., & Ropar, D. (2020). Neurotype-matching, but not being autistic, influences self and observer ratings of interpersonal rapport. *Frontiers in psychology*, 2961.
- DeBrabander, K. M., Morrison, K. E., Jones, D. R., Faso, D. J., Chmielewski, M., & Sasson, N. J. (2019). Do first impressions of autistic adults differ between autistic and nonautistic observers?. *Autism in Adulthood*, 1(4), 250-257.
- Fletcher-Watson, S., De Jaegher, H., Van Dijk, J., Frauenberger, C., Magnée, M., & Ye, J. (2018). Diversity computing. *Interactions*, 25(5), 28-33.
- Frisby, B. N., and Martin, M. M. (2010). Instructor-student and student-student rapport in the classroom. *Commun. Educ.* 59, 146–164. doi: 10.1080/03634520903564362
- Rifai, O. M., Fletcher-Watson, S., Jiménez-Sánchez, L., & Crompton, C. J. (2022). Investigating markers of rapport in autistic and nonautistic interactions. *Autism in Adulthood*, 4(1), 3-11.
- Sasson, N. J., Faso, D. J., Nugent, J., Lovell, S., Kennedy, D. P., & Grossman, R. B. (2017). Neurotypical peers are less willing to interact with those with autism based on thin slice judgments. *Scientific reports*, 7(1), 1-10.
- Sheppard, E., Pillai, D., Wong, G. T. L., Ropar, D., & Mitchell, P. (2016). How easy is it to read the minds of people with autism spectrum disorder? *Journal of autism and developmental disorders*, 46(4), 1247-1254.

3. I suspect that the research will be replicated in part because any diversity introduced into a transmission chain will tend to degrade the original message, as the message requires more extensive translation into different frames of reference. One would expect the same results for chains that mixed nationality, age, or social class etc. In a sense, this transmission paradigm (with the outcome being accuracy) could be interpreted as a measure of the extent to which the members of the chain share assumptions, communication heuristics, and frames of reference. The research states that participants will be matched for age, gender, and IQ. I wonder if there should also be some matching for social class, education level, and linguistic ability.

Thank you for raising this interesting point. We agree that demonstrating a specific effect of neurotype will be more powerful if we can capture and match for as many other factors as possible.

To incorporate your recommendations relating to social class we will ask participants to provide details of parental education level (both as years of education, and highest educational level achieved), a proxy for socio-economic status (Aarø et al., 2019) which can be used to compare across participants in all groups at all sites.

Additionally, we will ask participants to provide details of their education level (both as years of education, and highest educational level achieved).

To measure IQ, we are using the Wechsler Abbreviated Scale of Intelligence II (WASI-II), which includes a verbal IQ measure. We will break down the matching by verbal and performance scores on the WASI-II to examine comparability between the groups specifically on linguistic ability.

We will use these variables to match groups as far as possible. If it is not possible to match groups, we will examine the possible impact of this by entering the variable as a co-variate in a post-hoc analysis

We have now included the following information in-text (page 6) relating to this change:

We will attempt to ensure that groups are comparable on gender, age, educational level, linguistic ability, IQ, and socio-economic status (as indicated by parent education level (Aarø et al., 2019)). If it is not possible to match groups, we will examine the possible impact of this by entering the variable as a co-variate in a post-hoc analysis.

New references added

Aarø, L. E., Flisher, A. J., Kaaya, S., Onya, H., Namisi, F. S., & Wubs, A. (2009). Parental education as an indicator of socioeconomic status: improving quality of data by requiring consistency across measurement occasions. *Scandinavian journal of public health*, 37(2_suppl), 16-27.

Reviewer #2:

Remarks to the Author:

In this Registered Report, the authors report on the use of a diffusion chain method to examine information transfer in groups of autistic people, groups of non-autistic people, and mixed groups of autistic and non-autistic people. The research questions are important and relevant to a broad, multidisciplinary audience, and the proposed hypotheses are logic and plausible. The methodology and analysis pipeline (including statistical power analysis) are appropriate and provided in sufficient detail for replica.

Suggestions for improvement:

1) Transmission of information. As in Crompton et al., 2020, “a participant’s final score will correspond to the number of details they recall when recounting the passage to the next person in the chain, out of a maximum of 30”. Recall (number of details transmitted out of a maximum of 30) measures the quantity of information transmitted to the next person. In addition, the authors may want to measure the fidelity (accuracy) with which the message content is transmitted. This could be done, for example, by calculating the number of distortions and the number of new details that are generated. An example of this approach can be found in: <https://doi.org/10.1073/pnas.1421883112>

Thank you for your review and for the suggestions you have made to improve our manuscript.

We agree that examining the number of distortions/new details that occur within chains could be an interesting way to look at information transfer within groups, and appreciate this suggestion. As we have no way to predict what the distribution of this variable may be, we have included this within the Exploratory Analysis section. We have proposed formulating the variable, checking the distribution, and if it is sufficient for analysis, we will repeat a similar analysis to that for Hypothesis 1. We have now included the following detail on page 15 outlining this new proposed analysis:

Exploring Distortions in Diffusion Chain Content: In diffusion chains, the content of an original piece of information generally degrades over repeated social transmissions (Flynn & Whiten, 2008). However, it is also possible that information becomes distorted, with participants adding detail not included in the original information (e.g. “She turned left at the blue windmill” becomes “She turned left at the blue flowery windmill”) or making an error in information transfer (e.g. “She turned left at the blue windmill” becomes “She turned left at the blue castle”), which is then transferred to subsequent participants (Moussaïd, Brighton & Gaissmaier, 2015). Thus, in addition to counting the number of correct details transmitted, we will also examine transcripts of interactions to quantify distortions occurring in information transfer. We will examine and visualise the data and check the assumptions of the analysis method as described in Hypotheses 1. If the assumptions are met, we will perform a linear regression analysis, using a simple linear model, and if not, we will apply Bayesian analysis

methods. This will include the dependent variable of individual participant distortion score (i.e. how many distortions each participant introduced), and predictor variables of chain type (autistic, non-autistic, mixed) and chain position (1 to 6). Collectively, analyses will therefore determine both if information degrades differently across chain-types and whether new incorrect information is generated to a greater degree in some chain types (and by some participants) compared to others.

New references

Moussaïd, M., Brighton, H., & Gaissmaier, W. (2015). The amplification of risk in experimental diffusion chains. *Proceedings of the National Academy of Sciences*, *112*(18), 5631-5636.

2) Mismatch model of ASD. The authors derive their predictions from a mismatch-model of ASD's difficulties in social interaction. This model has received empirical support from studies showing that kinematic similarity is important for action prediction. ASD observers are better able to read/predict ASD actions than TD actions (<https://doi.org/10.1073/pnas.2114648119>) and conversely, TD observers are better able to read/predict TD actions than ASD actions (<https://doi.org/10.1073/pnas.2114648119>; <https://doi.org/10.1073/pnas.2011311117>).

The authors may want to refer to these studies as supporting a kinematic dissimilarity hypothesis of ASD difficulties in social interaction (<https://doi.org/10.1098/rstb.2015.0372>).

In this view, the difficulties in social interaction would not arise because of social judgements, and misunderstandings of non-autistic social partners (a "deficit" on the side of non-autistic social partners) but because of objective differences in the way TD and ASD communicate.

Thank you so much for this suggestion. We agree that it may be of interest to our readers to include the kinematic dissimilarity theory as a proposed mechanism of interactive differences between autistic and non-autistic people. We have incorporated your suggested literature in the introduction as follows:

"This is supported by empirical studies of the kinematic dissimilarity hypothesis of social interaction, which suggests that kinematic similarity is important for action prediction and social interaction (Cook, 2016). There are well documented kinematic differences between autistic and non-autistic people (Cook, Blakemore & Press, 2013), and recent studies have found that autistic observers are more able to accurately predict autistic actions than non-autistic actions (Montobbio, et al., 2022), and conversely non-autistic observers are more able to accurately predict non-autistic actions than autistic actions (Cook, 2016). These data suggest that difficulties in social interactions arise because of objective differences in the way that autistic and non-autistic people communicate, rather than an autistic social deficit".

New references

Cook, J. L., Blakemore, S. J., & Press, C. (2013). Atypical basic movement kinematics in autism spectrum conditions. *Brain*, *136*(9), 2816-2824.

Cook, J. (2016). From movement kinematics to social cognition: the case of autism. *Philosophical Transactions of the Royal Society B: Biological Sciences*, *371*(1693), 20150372.

Montobbio, N., Cavallo, A., Albergo, D., Ansuini, C., Battaglia, F., Podda, J., ... & Becchio, C. (2022). Intersecting kinematic encoding and readout of intention in autism. *Proceedings of the National Academy of Sciences*, *119*(5), e2114648119.

Response to additional Editor Feedback – received 15th May 2022.

1) Provide a sampling plan based on a power-analysis that complies with our guidelines. It is unclear how the effect size was chosen. There should be a justification for why the targeted effect size is the smallest effect of theoretical or practical significance, for which $\eta^2 = 0.24$ appears a very large value.

A mixed design for chains with chain type as the between factor and position as the within factor increases the power of the study relative to a between-subjects design. We modified the linear model so that each chain (rather than participant) is treated as an independent observation and proportion of

recalled information from a participant in a chain is considered as a repeated measure, allowing for dependencies between participants within a chain. To test the feasibility of this approach, we applied a corresponding linear model with repeated measurements to the original data (Crompton et al., 2020), It produced larger effect sizes for the main effects and interaction: chain type partial $\eta^2=0.52$, position in chain 0.87, interaction 0.20. For equivalent analyses of rapport scores we found a partial η^2 of 0.19 for the interaction. Assuming a statistical significance level of $p=0.05$, a medium effect size of $\eta^2=0.06$, a correlation of $r=0.4$ and non-sphericity with a correction of $\epsilon =0.7$ (Greenhouse-Geisser) an a priori power analysis (G*Power 3.1; Faul et al., 2009) suggests 54 chains or $54 \times 6 =324$ participants for the present study.

We have uploaded our R-code and dataset that this power simulation is based on to the OSF, to allow reviewers to view this in case there are further questions about details. These files are available at https://osf.io/3hnfj/?view_only=ffd597be20dc4d7488eb1c8f2b3a7d00

We have made additions to the *Sampling Plan* section of the manuscript (marked in red, pages 9-11) which we hope will provide sufficient clarity on our process.

- 2) Respond to the referees' concerns by implementing the necessary changes in the protocol, rather than rebutting their points of critique without undertaking any changes to the design.

As noted above, we strongly feel that self-rated rapport is a more appropriate measure for the aims of this study than observer-rated rapport. Normative external indicators of rapport are less likely to be observed between autistic pairs and may be undetected or misinterpreted by observers. This means that even if autistic pairs are experiencing high rapport (as we would expect, based on the previous study), external observers are likely to rate them as having low rapport (based on the neurotypical lens of external rapport measures).

The central aim of this study is to examine whether participants experience rapport with their partner, and if that feeling is mutual. This also replicates the procedure from the original study.

Observer perceptions of rapport are a separate research question that we could subsequently address with the data we collect but are not central to the core aims of this specific study. It would also add a significant time and resource commitment that would slow down completion of the study.

We feel we have provided a thorough explanation to the reviewer about why self-rated rapport is a more appropriate measure to use in this study. We have also now included some of this explanation in the paper itself (pages 8-9), with additions shown in red.

However, as stated in our response above, if the reviewer is not persuaded by our rebuttal, we are prepared to change the design in a future revision.

- 3) Do not remove hypotheses from the manuscript. Instead, the work must be sufficiently powered (95%) to test the relevant hypotheses.

We have now included all hypotheses that were in the original protocol.

Thank you so much for taking the time to seek and provide expert statistical feedback on this manuscript. As R4 notes, the statistical planning for this project is complicated by the within / between manipulations associated with the diffusion chain design and we are grateful for this opportunity for scrutiny and improvement, as well as for the support of the reviewer for our plans.

We also note the many encouraging comments from Reviewers 2 and 3 and thank them for their time and approval of changes already made.

Please find attached a revised manuscript with changes in response to reviewers marked like this, and a number of corrections to formatting as per the checklist provided by the editor.

Reviewer #4: Remarks to the Author

I was tasked with evaluating the report's sample size and power analysis. The authors need to update a few things, but with such a large sample size, they will likely be sufficiently powered for finding medium effects.

Thank you for this support of our research design and analysis plan.

1. The sample size seems appropriate, but clarification is necessary about the model chosen for the simulation-based power analysis. Here, the authors are using an intercept-only model ("1 | chain"), but no theoretical justification for the random effects structure is given. Choosing intercept-only models out-of-the-box, without a strong theoretical reason, can inflate the probability of alpha errors and can reduce generalisability:

Barr DJ, Levy R, Scheepers C, Tily HJ. Random effects structure for confirmatory hypothesis testing: Keep it maximal. *J Mem Lang.* 2013 Apr;68(3):10.1016/j.jml.2012.11.001.

Either choosing the "maximal" random effect structure (probably "1 + position | chain" if I understood the within/between manipulations correctly) is necessary, or following very careful and conservative random effects-simplification is required:

Bates D., Kliegl R., Vasishth S., & Baayen H. Parsimonious mixed models. arXiv. 2015; (arXiv:1506.04967v2) <https://arxiv.org/abs/1506.04967>

Either way, this needs to be justified and/or the power analysis needs to be re-run with the justified random-effects structure. Most likely the selected sample size will nevertheless be sufficiently large.

The simulation-based power analysis uses the original data by Crompton et al. (2020) with 8 participants in each of the 3 chains of each condition (N=72). This is the minimum number of chains required to conduct an analysis with random intercepts. If 'position' is included as another random effect then the model fails to converge and we can no longer run a simulation-based power analysis in simr.

Increasing the sample size to N=324, with 6 participants in each of the 18 chains of each condition, should give us the opportunity to fit the maximal model (Barr et al., 2013) and/or to identify the most parsimonious model (Bates et al., 2015; James et al. 2014).

We have made changes and inserted text based on the explanation above into the manuscript on pages 9 and 10.

2. The authors state that the simulations “are based on reduced coefficients”, but it is not transparent how much they were reduced. Providing a % would be important.

The estimated coefficients were reduced by one-third to make sample size estimation more conservative. An amendment has been made in the text near the bottom of page 10: “... *because they are based on reduced coefficients.*” has been replaced by “... *because they are based on the estimate coefficients reduced by one-third.*” This information is also provided in the caption to Figure 2.

3. Finally, the OSF link (<https://osf.io/us9c7/>) did not lead to R code. The authors state “Further details of the power analysis and simulations in R can be found in the R-file”, so this needs to be updated.

Apologies for this oversight. The file PowerAnalysis.R has now been uploaded to <https://osf.io/us9c7/> .